EMBO
Molecular Medicine

# Glycolysis downregulation is a hallmark of HIV-1 latency and sensitizes infected cells to oxidative stress

Iart Luca Shytaj[1,2,3,*], Francesco Andrea Procopio[4], Mohammad Tarek[5] , Irene Carlon-Andres[6,7,8],
Hsin-Yao Tang[9] , Aaron R Goldman[9], MohamedHusen Munshi[10] , Virender Kumar Pal[10],
Mattia Forcato[11] , Sheetal Sreeram[12] , Konstantin Leskov[12] , Fengchun Ye[12] , Bojana Lucic[2,13] ,
Nicolly Cruz[3], Lishomwa C Ndhlovu[14], Silvio Bicciato[11], Sergi Padilla-Parra[6,7,8] , Ricardo Sobhie Diaz[3],
Amit Singh[10] , Marina Lusic[2,13] , Jonathan Karn[12] , David Alvarez-Carbonell[12,†] &
Andrea Savarino[1,**,†]

## Abstract

HIV-1 infects lymphoid and myeloid cells, which can harbor a latent proviral reservoir responsible for maintaining lifelong infection. Glycolytic metabolism has been identified as a determinant of susceptibility to HIV-1 infection, but its role in the development and maintenance of HIV-1 latency has not been elucidated. By combining transcriptomic, proteomic, and metabolomic analyses, we here show that transition to latent HIV-1 infection downregulates glycolysis, while viral reactivation by conventional stimuli reverts this effect. Decreased glycolytic output in latently infected cells is associated with downregulation of $NAD^+$/NADH. Consequently, infected cells rely on the parallel pentose phosphate pathway and its main product, NADPH, fueling antioxidant pathways maintaining HIV-1 latency. Of note, blocking NADPH downstream effectors, thioredoxin and glutathione, favors HIV-1 reactivation from latency in lymphoid and myeloid cellular models. This provides a "shock and kill effect" decreasing proviral DNA in cells from people living with HIV/AIDS. Overall, our data show that downmodulation of glycolysis is a metabolic signature of HIV-1 latency that can be exploited to target latently infected cells with eradication strategies.

**Keywords** glycolysis; HIV-1 latency; oxidative stress; pentose cycle; pyrimidine metabolism
**Subject Categories** Metabolism; Microbiology, Virology & Host Pathogen Interaction

## Introduction

Decades after its outbreak, the HIV/AIDS pandemic remains one of the main causes of morbidity and mortality of humankind, leading to almost one million victims per year (source: UNAIDS). Moreover, due to the severe medical and economic crisis caused by coronavirus disease 2019 (COVID-19), adherence to and availability of antiretroviral therapies (ART) are decreasing in areas where HIV/AIDS prevalence is particularly high (Jewell *et al*, 2020; Jiang *et al*, 2020). This worsening of the death toll highlights the fragility of current therapeutic approaches, based on lifelong ART administration. Therefore, a cure for HIV/AIDS is an unmet medical need of

1 Department of Infectious Diseases, Italian Institute of Health, Rome, Italy
2 Department of Infectious Diseases, Heidelberg University Hospital, Heidelberg, Germany
3 Infectious Diseases Department, Federal University of São Paulo, São Paulo, Brazil
4 Service of Immunology and Allergy, Lausanne University Hospital, University of Lausanne, Lausanne, Switzerland
5 Bioinformatics Department, Armed Forces College of Medicine (AFCM), Cairo, Egypt
6 Division of Structural Biology, Wellcome Centre for Human Genetics, University of Oxford, Oxford, UK
7 Department of Infectious Diseases, Faculty of Life Sciences & Medicine, King's College London, London, UK
8 Randall Division of Cell and Molecular Biophysics, King's College London, London, UK
9 The Wistar Institute, Philadelphia, PA, USA
10 Indian Institute of Science, Bangalore, India
11 Department of Life Sciences, University of Modena and Reggio Emilia, Modena, Italy
12 Department of Molecular Biology and Microbiology, Case Western Reserve University, Cleveland, OH, USA
13 German Center for Infection Research, Heidelberg, Germany
14 Division of Infectious Diseases, Department of Medicine, Weill Cornell Medicine, New York, NY, USA
*Corresponding author. Tel: +55 11 5576 4834; E-mail: shytaj.luca@unifesp.br
**Corresponding author. Tel: +39 06 4990 2305; E-mail: andrea.savarino@iss.it
†These authors contributed equally to this work

growing importance for people living with HIV/AIDS (PLWH). The quest for a cure is hampered by persistence of transcriptionally silent proviruses within latently infected cells, which render these cells hard to discriminate from their uninfected counterparts (Finzi et al, 1999). Latent HIV-1 DNA can be mainly found in viral reservoirs such as CD4$^+$ T cells (Van Lint et al, 2013); however, myeloid cells (in particular microglia) can also contribute to persistence of the infection during ART (Sattentau & Stevenson, 2016). Pinpointing molecular features to allow selective targeting of latently HIV-1-infected cells would represent a significant, and perhaps decisive, step in the quest for an HIV/AIDS cure.

A possible approach to reach this goal is the investigation of the metabolic pathways exploited by the retrovirus to actively replicate and enter a latent state. Several metabolic pathways have so far been explored in the quest of a cure for HIV/AIDS. In the context of a retroviral infection (characterized by a DNA proviral phase), one of the first metabolic pathways to be explored was nucleotide metabolism (Declercq E. New acquisitions in the development of anti-HIV agents, 1989), in turn linked to several other metabolic pathways, including oxidative stress (Seifert et al, 1989; De Clercq, 2011), one of the most investigated but not yet fully understood pathways. In particular, inhibition of oxidative stress has been examined, starting from last century's Nineties (Garaci et al, 1997), as a strategy to induce a so-called "block and lock" effect, sending the provirus to a deep latency state that is not reactivated when ART is interrupted (Singh et al, 2021). Conversely, the activation of oxidative stress has been explored in the context of the so-called "shock and kill" strategies, aimed at purging the proviral reservoirs by inducing HIV-1 escape from latency and their consequent elimination by cytopathogenicity or by the immune system (Yang et al, 2019). Profoundly linked to oxidative stress, iron metabolism has been shown to lead, through its manipulation, to the selective death of infected cells (Savarino et al, 1999; Hanauske-Abel et al, 2013; Shytaj et al, 2020), but its clinical applications have so far been hampered by off-target effects. Other pathways that might have therapeutic potential in the setting of HIV-1 infection are autophagy (Loucif et al, 2021) and the intertwined tryptophan/kynurenine (reviewed in: Routy et al, 2015) and nicotinamide metabolism (Savarino et al, 1997; Lebouché et al, 2020). The importance of understanding the relationship between cell metabolism and the infection is emphasized by a recent case of long-term control of HIV-1 without ART (Oral Abstracts from the 23rd International AIDS Conference, 2020). This individual had been treated with a combination of intensified ART and nicotinamide, although the anecdotal character of this case is suggested by lack of success in other similar clinical attempts (Lebouché et al, 2020). Cells infected with HIV-1 display decreased levels of NAD$^+$, an important substrate of glycolytic reactions (Murray et al, 1995), and supplementation of the NAD$^+$ precursor nicotinamide influences viability of productively HIV-1-infected cells (Savarino et al, 1996).

In this regard, it is interesting to point out that the cells more susceptible to HIV-1 infection are characterized by an increased glycolytic rate (Valle-Casuso et al, 2019). Increased glycolysis is linked to cellular activation, which is necessary for active HIV-1 replication; of note, in CD4$^+$ T cells and monocytes of HIV-1-infected individuals, glucose consumption is increased and the expression of the glucose transporter GLUT-1 is upregulated (Palmer et al, 2014a, 2014b). Conversely, reports on the effect of HIV-1 infection on glycolysis have been sparse and, in some respects, conflicting. After ART implementation, PLWH showed to be more susceptible to hyperglycemia than the general population (Dubé et al, 1997). On the other hand, upregulated glucose metabolism was reported to favor apoptosis of infected CD4$^+$ T cells (Hegedus et al, 2014). Although these combined sets of data strongly support a role of active glycolysis in determining susceptibility to HIV-1 infection as well as an altered glucose metabolism in PLWH, the role of glycolysis in viral latency, and therefore in possible curing strategies, has as yet remained unclear.

Indirect evidence of an interplay between glycolysis and HIV-1 latency can also be drawn by the dysregulation of redox pathways, which are intertwined with glycolytic metabolism in several cell types and pathological conditions (Kondoh et al, 2007; Locasale & Cantley, 2011). This interconnection might be relevant, as we recently showed that HIV-1 infection leads to enhancement of antioxidant defenses in primary CD4$^+$ T cells (Shytaj et al, 2020). HIV-1 infection causes an initial oxidative stress (Daussy et al, 2020), which then leads to the nuclear translocation of the master antioxidant transcription factor Nrf2. This translocation in turn induces transcription of several proteins involved in antioxidant response, including glucose-6-phosphate dehydrogenase (G6PDH), which diverts glucose-6-phosphate from the glycolytic pathway to the pentose cycle, responsible for the production of the antioxidant NADPH. Therefore, elucidating the specific glycolysis/redox state interconnection during HIV-1 infection would be of pivotal importance for understanding the molecular events which lead infected cells to either die or establish a latent infection.

Herein, we combine transcriptomic, proteomic, and metabolomic data sets, including single-cell analyses, to show that, during transition to HIV-1 latency, glycolysis is downregulated. In line with this, our results show that latently infected cells able to undergo HIV-1 reactivation are more responsive to glycolysis reactivation. Moreover, we show that downregulation of glycolysis in latently infected cells is accompanied by higher reliance on the antioxidant thioredoxin (Trx) and glutathione (GSH) systems for cell survival. Our results highlight the possibility to exploit glycolytic imbalances induced by HIV-1 infection for the elimination of retrovirally infected cells. This result may improve our knowledge of pathways that can be targeted by strategies aimed at eradicating HIV/AIDS.

# Results

## CD4$^+$ T cells progressively downregulate the expression of glycolytic enzymes in response to HIV-1 infection

To study transcriptomic profiles upon infection, we used microarray and RNA-Seq data sets generated in primary CD4$^+$ T cells infected with HIV-1$_{pNL4-3}$. The cellular model employed is based on longitudinal sample collection to cover different time points, from early HIV-1 infection [day 3 post-infection (p.i.)] to peak retroviral replication (days 7–9 p.i.) and latency/survival of infected cells (day 14 p.i.). The detailed features and validations of this cellular model have been described previously elsewhere (Shytaj et al, 2020).

Microarray results, comprising a mean of late time points of infection (days 7–14 p.i), highlighted significant downregulation of the glycolytic pathway in infected cell cultures, which were

independently evidenced by Gene Set Enrichment Analysis (GSEA; Subramanian *et al*, 2005) either using a customized gene set comprising enzymes involved in human glycolysis (henceforth, HUMAN-GLYCOLYSIS) or using the Reactome or Biocarta databases (Fig 1A; Appendix Table S1). Entrance of the infected cells into a hypometabolic state was also shown by downregulation of glycolysis-independent transcriptional and translational pathways, although downregulation of these pathways did not reach the same level of convergence among the databases examined as compared to the glycolytic pathway (Appendix Table S1). Moreover, among the

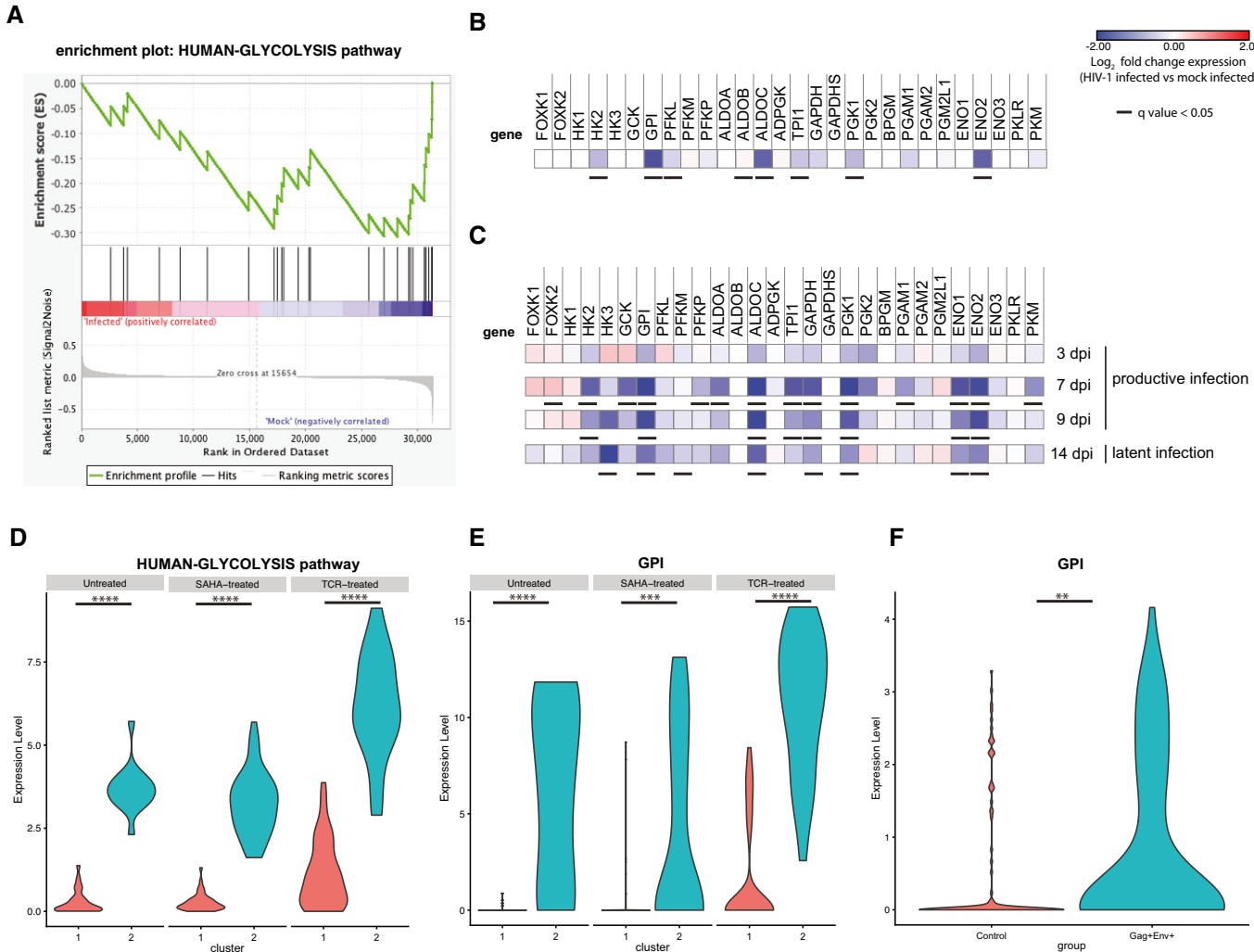

**Figure 1. Transcriptional modulation of glycolytic enzymes in productively or latently HIV-1-infected cells.**

A–C  Primary CD4[+] T cells were activated with α-CD3-CD28 beads and infected with HIV-1$_{pNL4-3}$ or mock-infected. Cells were cultured for 2 weeks post-infection (p.i.) to model different infection stages (days 3–9 p.i., i.e., productive infection; day 14 p.i., i.e., latent infection) and subjected to microarray (A, B; *n* = 2) or RNA-Seq (C; *n* = 3) analysis. (A) Gene set enrichment analysis (GSEA) of the expression of the glycolytic pathway (HUMAN-GLYCOLYSIS) in mock-infected or HIV-1-infected cells. (B,C) Heatmaps of the relative expression of glycolytic enzymes upon HIV-1 infection. Data are expressed as Log$_2$ fold change in HIV-1-infected vs mock-infected cells. For microarray data (B), expression values of infected and mock-infected cells at different time points were pooled. For RNA-Seq data (C), expression values in infected cells were normalized using the corresponding time point in mock-infected cells. Adjusted *P*-values (q values) to account for multiple testing were calculated by Significance Analysis of Microarrays [SAM (Tusher *et al*, 2001)] and Deseq2 for RNA-Seq data (Love *et al*, 2014).

D–F  scRNA-Seq of the expression of the entire glycolytic pathway or of glucose phosphate isomerase only (*GPI*) in primary CD4[+] T cells infected *in vitro* (D,E) or CD4[+] T cells of PLWH (F). In panels (D, E), cells were infected with VSVG-HIV-1-GFP and sorted for viral expression as detailed in Golumbeanu *et al* (2018). Following latency establishment, cells were left untreated or HIV-1 expression was reactivated through suberoylanilide hydroxamic acid (SAHA) or α-CD3-CD28 engagement. Clusters 1 and 2 were identified by principal component analysis as described in Golumbeanu *et al* (2018). In panel (F), CD4[+] T cells were isolated from total blood of PLWH under ART as described in Cohn *et al* (2018). Viral expression was reactivated by treatment with phytohemagglutinin (PHA), and cells were sorted using antibodies against Env and Gag. Sorted cells were then subjected to scRNA-Seq analysis. The expression level of the HUMAN-GLYCOLYSIS pathway in (D) was calculated as the average expression of genes comprising the gene list; expression levels in clusters 1 and 2 were compared using Wilcoxon rank-sum test. For panels (E, F), significance of *GPI* differential expression level between clusters (E) or between control and Env[+] Gag[+] conditions (F) was assessed by the Wilcoxon rank-sum test encoded in FindMarkers Seurat R function. **$P < 0.01$, ***$P < 0.001$; ***$P < 0.0001$. For panels (D, E) *n* = 1 donor, 43 cells (untreated), 90 cells (SAHA), 91 cells (TCR), for panel (F) *n* = 3 donors, 109 cells (control) and 85 cells (Gag[+] Env[+]).

pathways more heavily perturbed by HIV-1 infection, there was the interferon pathway (Appendix Table S1), as expected, and pathways associated with both apoptosis and cell cycle, in line with the fact that, in an HIV-1-infected cell culture, some cells succumb to infection and others survive developing a proviral latent state.

We then analyzed in detail the glycolytic enzymes of the pathway HUMAN-GLYCOLYSIS (Fig 1B). The glycolytic enzymes characterized by significantly downregulated transcription upon infection were hexokinase 2 (*HK2*), glucose-6-phosphate isomerase (*GPI*), phosphofructokinase liver type (*PFKL*), aldolase fructose-bisphosphate C (*ALDOC*), triosephosphate isomerase 1 (*TPI1*), and enolase 2 (*ENO2*), thus suggesting a broad downregulation of the glycolytic pathway (Fig 1B). In particular, the highly significant downregulation of *GPI* suggests a glycolysis-specific effect, as this enzyme commits metabolites to the glycolytic, rather than to the alternative pentose phosphate pathway.

To expand these analyses, we explored an RNA-Seq data set derived from the same primary CD4$^+$ T-cell model, which was previously published elsewhere (Shytaj *et al*, 2020). In this data set, the number of donors and time points was higher, allowing the study of the expression of glycolytic enzymes during each infection stage (Fig 1C). In line with the microarray results, differential gene expression analysis of RNA-Seq data (DESeq2) (Love *et al*, 2014) highlighted significant downregulation of glycolysis in infected cell cultures, which were initiated during the late stages of HIV-1 replication (7–9 days p.i.) and persisted after retroviral replication had ceased (14 days p.i.) (Fig 1D). In line with our microarray analysis, enzyme transcriptional downregulation covered the main steps of glycolysis, and the RNA-Seq data set further suggested that various isoforms, in particular of *PFK*, might contribute differently to glycolysis downregulation during productive or latent infection [*PFK*-platelet (P); adjusted *P*-value = 0.04 at 7 days p.i. *PFK*-muscle (M); adjusted *P*-value = 0.04 at 14 days p.i., respectively]. Finally, analysis of a previously published proteomic data set of the same CD4$^+$ T-cell model (Shytaj *et al*, 2020) further corroborated the downregulating effect of HIV-1 on glycolysis, confirming significant downmodulation of GPI, PGK1, and TPI1 (Appendix Fig S1).

Overall, these data show that HIV-1 infection, during its transition to latency, is associated with downregulated expression of glycolytic genes.

## Expression of glycolytic enzymes is required for HIV-1 escape from latency in lymphoid and myeloid cells

The aforementioned results prove that glycolysis downregulation is initiated during productive infection and accompanies HIV-1 latency establishment. We then proceeded to specifically investigate the transcriptional regulation of the HUMAN-GLYCOLYSIS pathway upon the reverse process, i.e., reactivation from latency.

To this aim, we first analyzed two single-cell RNA-Seq (scRNA-Seq) data sets independently published by the groups of Ciuffi and Nussenzweig, respectively (Cohn *et al*, 2018; Golumbeanu *et al*, 2018). In the first data set, primary CD4$^+$ T cells infected with pseudotyped HIV-1-GFP/VSVG had been sorted according to GFP expression and allowed to revert to latency (Golumbeanu *et al*, 2018). Latently infected cells had then been either left untreated or subjected to HIV-1 reactivation by strong (α-CD3/CD28 antibodies) or weak stimuli [suberoylanilide hydroxamic acid (SAHA)].

Eventually, the transcriptomic profile was analyzed by scRNA-Seq (Golumbeanu *et al*, 2018). Using principal component analysis, the authors identified two cell clusters, which were less (cluster 1) or more (cluster 2) susceptible to HIV-1 reactivation (Golumbeanu *et al*, 2018). We analyzed the expression of the enzymes of the glycolytic pathway in both clusters and found that glycolytic enzymes were downregulated in cells of cluster 1 (Fig 1D). Interestingly, this relative downregulation, already visible in basal conditions, was maintained upon treatment with either anti-CD3/CD28 antibodies or SAHA (Fig 1D). Moreover, *GPI* expression was lower in the cell subpopulation less responsive to HIV-1 reactivation as compared to the more susceptible cell subpopulation (Fig 1E), in line with the results obtained on our model of latency establishment (Fig 1B and C). Of note, also when CD4$^+$ T-cell subsets were analyzed separately to account for potential differences in their metabolism (Loucif *et al*, 2020), cluster 1 was characterized by downregulation of glycolysis in each detectable subset, corroborating lowered glycolysis as a general marker of latent HIV-1 infection (Fig EV1).

To further validate the clinical relevance of these findings, we analyzed a second scRNA-Seq data set obtained from CD4$^+$ T cells of PLWH under ART. To generate this data set, the Nussenzweig group separated cells responsive to HIV-1 reactivation by detecting Gag/Env expression upon treatment with the pan-lymphocyte activator, phytohemagglutinin (PHA) (Cohn *et al*, 2018). Despite its limitations (the lower number of infected cells as compared to the previously mentioned model and the likely presence of a mixed population of infected and uninfected cells), this data set had the advantage of capturing a viral reservoir generated under the pathophysiologic conditions occurring *in vivo* in a clinical setting. In line with the results shown by scRNA-Seq of *in vitro* infected cells, the cell population responsive to HIV-1 reactivation displayed significant upregulation of *GPI* (Fig 1F).

We then expanded our analysis to include microglial cells transformed with SV40 and latently infected with HIV-1 [i.e., HC69 cells (Alvarez-Carbonell *et al*, 2017; Garcia-Mesa *et al*, 2017)]. Microglia constitutes one main myeloid retroviral reservoir during ART and is largely responsible for HIV-1 persistence in the central nervous system (Churchill & Nath, 2013). Our bulk RNA-Seq HC69 data, along with data on its uninfected equivalent (C20 cells), allowed a comparison of reactivated *vs*. latent infection through stimulation of latently infected cells with tumor necrosis factor [TNF (Garcia-Mesa *et al*, 2017)]. When infected cells were compared to their uninfected counterparts incubated under similar conditions, results showed a clear pattern of *GPI* downregulation being more significant in latent than in productive infection and accompanied by a more pronounced downregulation of the early glycolytic enzymes in the former (Appendix Fig S2). Finally, scRNA-Seq profiling of HC69 cells highlighted a significantly positive correlation between baseline HIV-1 expression and key glycolytic enzymes, including *GPI*, in the subset of cells in which proviral transcription was detectable (Fig EV2A and B). Conversely, treatment with dexamethasone (DEXA), which is a known glycolysis inhibitor (Ma *et al*, 2013), decreased both the proportion of HC69 cells expressing the transcripts of the glycolytic pathway and the baseline percentage of cells expressing HIV-1 transcripts (Fig EV2C).

Taken together, these results show that low expression of early glycolytic enzymes, in particular *GPI*, is associated with HIV-1

latency maintenance in multiple cellular models and subtypes and that at least partial restoration of glycolysis is required for latency disruption.

### Decreased initial glycolytic metabolism during latent HIV-1 infection

To evaluate changes in the metabolic profiles of cells characterized by productive and latent HIV-1 infection, we subjected lymphoid (Jurkat T cells) and myeloid (HC69 microglia) cell models to LC-MS/MS metabolomic profiling (Figs 2 and EV3). These cell types

offer the advantage of providing data from a homogeneously infected population. Four conditions were analyzed: quiescent, activated, latently infected, and productively infected.

Metabolite enrichment analysis showed significant changes in the glycolysis/gluconeogenesis pathway in both latently and productively infected cells, with glycolysis being one of the top scoring pathways in both cell types analyzed (Figs 2A and B, and EV3A and B).

When compared to uninfected cells in terms of single metabolites, latently infected cells showed significantly decreased levels of the initial glycolytic metabolite, i.e., glucose-6-phosphate, as well as

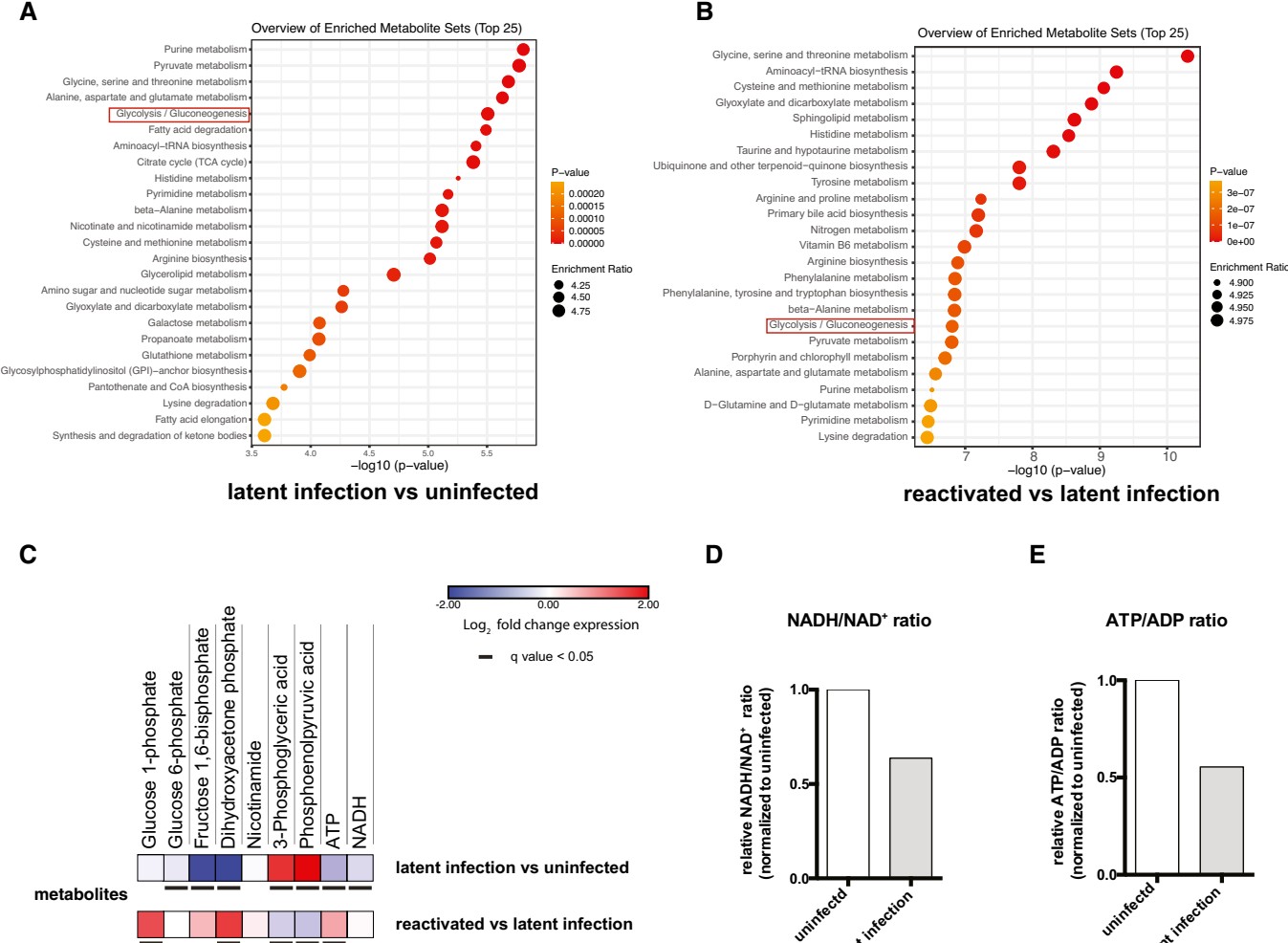

**Figure 2. Modulation of glycolysis and other metabolic pathways during productive or latent HIV-1 infection.**

Latently HIV-1-infected (2D10) or uninfected (E6) Jurkat T cells were subjected to metabolomic analysis under unstimulated conditions or following stimulation with TNF to reactivate latent HIV-1.

A, B  Metabolite enrichment analysis in latently infected cells as compared to their uninfected counterparts (A) and cells reactivated from latency and compared with latently infected cells (B). The top enriched pathways were ordered according to *P*-values obtained by Q statistics for metabolic data sets performed with Globaltest (MetaboAnalyst) (Xia *et al*, 2009).

C  Heatmaps of glycolytic metabolites in latently infected Jurkat T cells as compared to their uninfected counterparts or Jurkat T cells with HIV-1 reactivated by TNF as compared to latently infected cells. Data are displayed as Log$_2$ fold change expression. Adjusted *P*-values (*q* values) were calculated by the Benjamini–Hochberg false discovery rate.

D, E  Relative ratios of NADH/NAD$^+$ (D) and ATP/ADP (E) in latently infected and reactivated cells. Data were normalized using the matching uninfected control.

of the metabolite produced by PFK, i.e., fructose 1,6 bisphosphate (Figs 2C and EV3C). Since the PFK enzyme acts immediately downstream of GPI, downregulation of fructose 1,6 bisphosphate appears to be a consequence of the reduced expression of *GPI* in latently infected cells (as shown by both RNA-Seq and scRNA-Seq analyses: Fig 1C, E and F, Appendix Fig S2). When cells with reactivated HIV-1 were compared to latently infected cells, a specular result was obtained, with a relative increase in early glycolytic products (Figs 2C and EV3C), in line with the evidence that HIV-1 reactivation increases transcription of the glycolytic enzyme GPI (Figs 1E and F, and EV2B). The NADH/NAD$^+$ ratio was decreased in latent infection, in both lymphoid and myeloid cells (Figs 2D and EV3D). Moreover, the ATP/ADP ratio was also decreased in latently infected Jurkat T cells, but not in HC69 microglial cells (Figs 2E and EV3E), suggesting an alternative source of ATP, other than glycolysis, depending on the cell type examined. The late glycolytic metabolites, such as lactate, also showed variable trends during infection depending on the cell type analyzed (Figs 3 and EV4, Appendix Fig S3), suggesting that the downregulation of early glycolytic steps is a unifying metabolic signature of latency.

The variable regulation of late glycolytic metabolites can be explained by their supply from alternative and partially overlapping, or intertwined pathways. One such pathway is the pentose phosphate cycle, which can supply the glycolytic intermediate glyceraldehyde 3-phosphate through pentose conversion (Figs 3 and EV4). The activity of the pentose phosphate cycle in latently infected cells is shown by the increase in the pathway-specific metabolite sedoheptulose 7-phosphate (Figs 3 and EV4). However, NADPH levels in latently infected cells were altered depending on a general depression of nicotinamide metabolism (Figs 2A, 3 and EV4) and on the level of oxidized glutathione, which is generated by HIV-1 infection (Shytaj *et al*, 2020) and reduced by NADPH consumption (Benhar *et al*, 2016) (Figs 3 and EV4). Associated with the increased activity of the pentose phosphate pathway is the upregulated pyrimidine metabolism in latently infected cells (Figs 2A, 3 and EV4).

Apart from the pentose phosphate pathway, there was evidence of a general metabolic attenuation in latently infected cells, with an impaired lipidic catabolism in lymphoid and myeloid cells, respectively (Figs 2A, 3, EV3A and EV4). This was also in line with an increased number of lipid droplets in HIV-1-infected cells, although their density remained unchanged (Appendix Fig S4).

Finally, the Krebs cycle, which is the energy production pathway subsequent to glycolysis and lipid metabolism, also appeared to be less active in latently infected cells, as shown by the decrease in its main supply, i.e., acetyl-CoA (Fig 3). However, in latently infected myeloid cells (Fig EV4), the Krebs cycle was in part maintained by an alternative supply from glutamate metabolism (Figs EV3A and EV4), as previously shown by Castellano *et al* (2019) and Gupta *et al* (2010).

Overall, these results show that latently HIV-1-infected lymphoid and myeloid cells are characterized by downregulation of early glycolytic metabolites and by a compensatory activity of the parallel pentose phosphate pathway. These changes are less pronounced in productively infected cells, thus supporting their involvement in HIV-1 latency.

## HIV-1 transcription is a specific driver of metabolic changes

We then tested the hypothesis that HIV-1 expression may be a direct cause of metabolic alterations. To isolate the metabolic effects of the transcriptional status of HIV-1, we focused on the role of the lentiviral transactivator Tat protein, which is required for the expression of the integrated provirus through binding to the trans-activation response (TAR) region, but can also activate numerous cellular genes and molecular networks, especially those involved in inflammatory responses (Clark *et al*, 2017).

First, we examined the effect of exogenous Tat administration (i.e., known to lead to intracellular uptake of the fully active protein) (Albini *et al*, 1995; Perkins *et al*, 2008) using the U1 cell line, which displays a defective Tat/TAR axis (Emiliani *et al*, 1998). The choice of Tat was also due to the fact that this protein has a well-documented activity as oxidative stress inducer (Toborek *et al*, 2003; El-Amine *et al*, 2018) and oxidized glutathione was recognized as a main NADPH consumer in our metabolomic analysis (Fig ). Tat administration was associated with increased expression of HIV-1 (Appendix Fig S5A), as expected, as well as with upregulation of the glucose transporter 1 (Glut1) in infected—but not in uninfected—cells (Appendix Fig S5B and C), in line with the upregulation of glycolytic enzymes that we observed upon HIV-1 reactivation from latency (Fig 1). Administration of Tat also primed infected cells for nucleotide (pyrimidine) synthesis, as shown by the significant increase in carbamoyl-phosphate synthetase 2 (CAD) expression (Appendix Fig S5B). This increase was visible in both infected and uninfected cells (Appendix Fig S5B and C). These changes are in line with an initial upregulation of cellular metabolism necessary for HIV-1 replication, as reported by others (Valle-Casuso *et al*, 2019). On the other hand, glutamine fructose 6-phosphate transaminase (GFPT1), an enzyme starting a pathway

**Figure 3.  Schematic depiction of glycolysis and related metabolic networks in latently HIV-1-infected Jurkat T cells.**

The regulation of metabolic pathways was reconstructed based on the metabolomic data of latently infected Jurkat T cells (2D10) as compared to their uninfected counterpart (E6). Enzymes are shown in italics. The figure includes the main energetic pathways described in the paper, as well as relevant connections with pathways that were significantly enriched in the analysis shown in Fig 2A (gray boxes). The red arrows indicate the proposed path of the glucose carbon chains in latently infected cells. Solid lines indicate direct connections. Dashed lines indicate indirect connections involving intermediate metabolites not shown in the figure. Networks were built using the Cytoscape software (http://www.cytoscape.org) and the MetScape plugin (http://metscape.ncibi.org/tryplugin.html) (Gao *et al*, 2010) and adapted using Adobe Illustrator (v 16.03). ACLY = ATP citrate lyase; ACO = aconitase; ADPGK = ADP-dependent glucokinase; ALDO = fructose 1,6 bisphosphate aldolase; ENO1 = enolase 1; FBP1 = fructose-bisphosphatase; PFK = phosphofructokinase 1; FH = fumarate hydratase; H6PD = hexose-6-phosphate dehydrogenase; HK1 = hexokinase 1; G6PD = glucose-6-phosphate dehydrogenase; GAPDH = glyceraldehyde 3-phosphate dehydrogenase; GCK = glucokinase; GPDH = glycerol 3-phosphate dehydrogenase; GPI = glucose-6-phosphate isomerase; KORA = ketoglutarate dehydrogenase A; IDH = isocitrate dehydrogenase; LDHAL6A = lactate dehydrogenase A like 6A; MDH1 = malate dehydrogenase 1; PCK1 phosphoenolpyruvate carboxykinase 1; PFK = phosphofructokinase; PGAM4 = phosphoglycerate mutase family member 4; PGD = 6-phosphogluconate dehydrogenase; PGK1 = phosphoglycerate kinase 1; PGLS = 6-phosphogluconolactonase; PGM1 = phosphoglucomutase 1; PKLR = pyruvate kinase L/R; PRPS1 = ribose phosphate pyrophosphokinase 1; RPIA = ribose 5-phosphate isomerase; SDHA = succinate dehydrogenase complex flavoprotein subunit A; SUCD = succinate semialdehyde dehydrogenase; TKT = transketolase; TPI = triose phosphate isomerase.

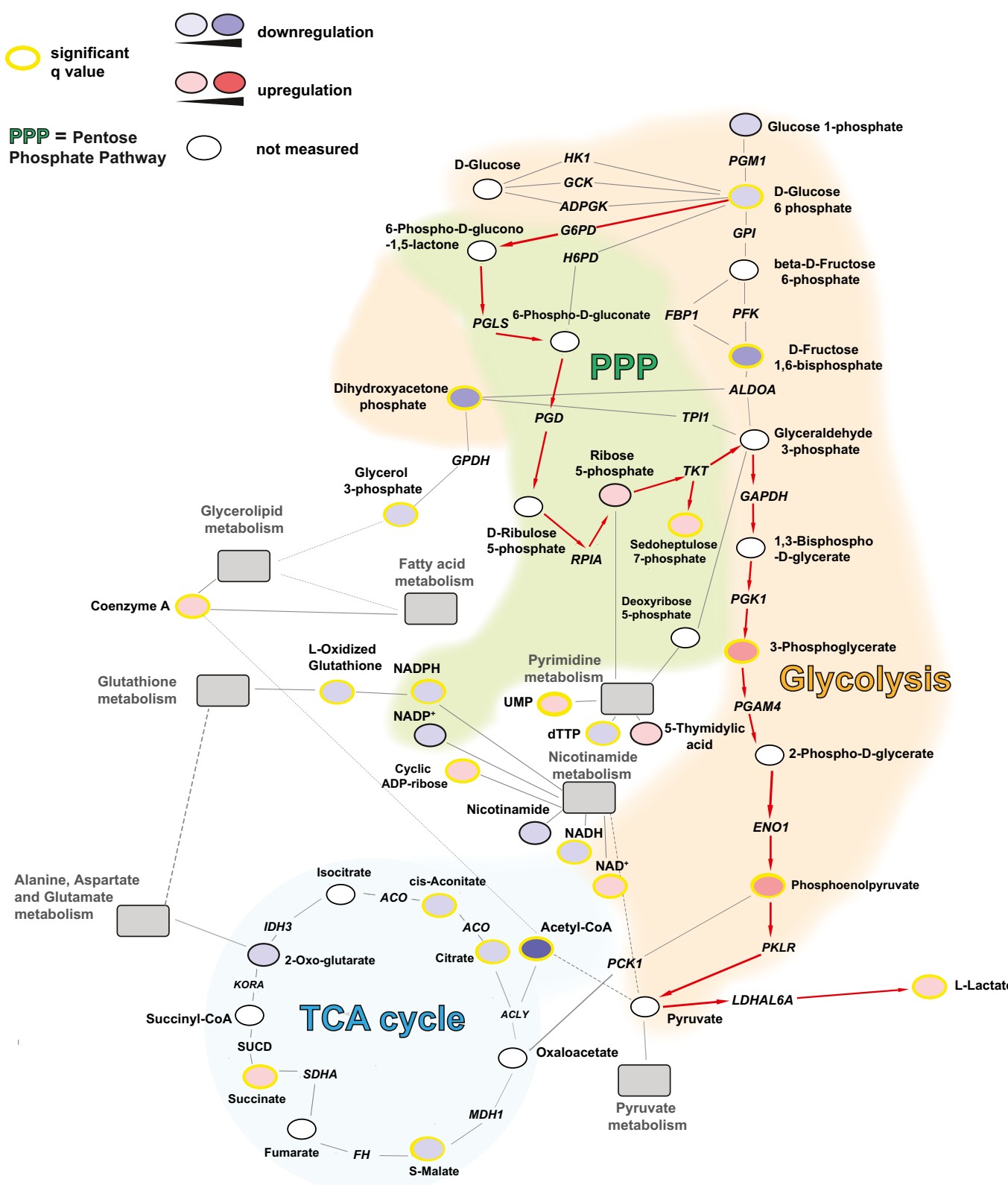

**Figure 3.**

bridging the pyrimidine and glutamine metabolism (Yelamanchi *et al*, 2016), was not modulated by the presence of Tat (Appendix Fig S5B and C).

Among other pathways affected by HIV-1 infection, there was a trend toward an increase in G6PD, which is a rate-limiting enzyme of the pentose phosphate pathway (Appendix Fig S5B and C).

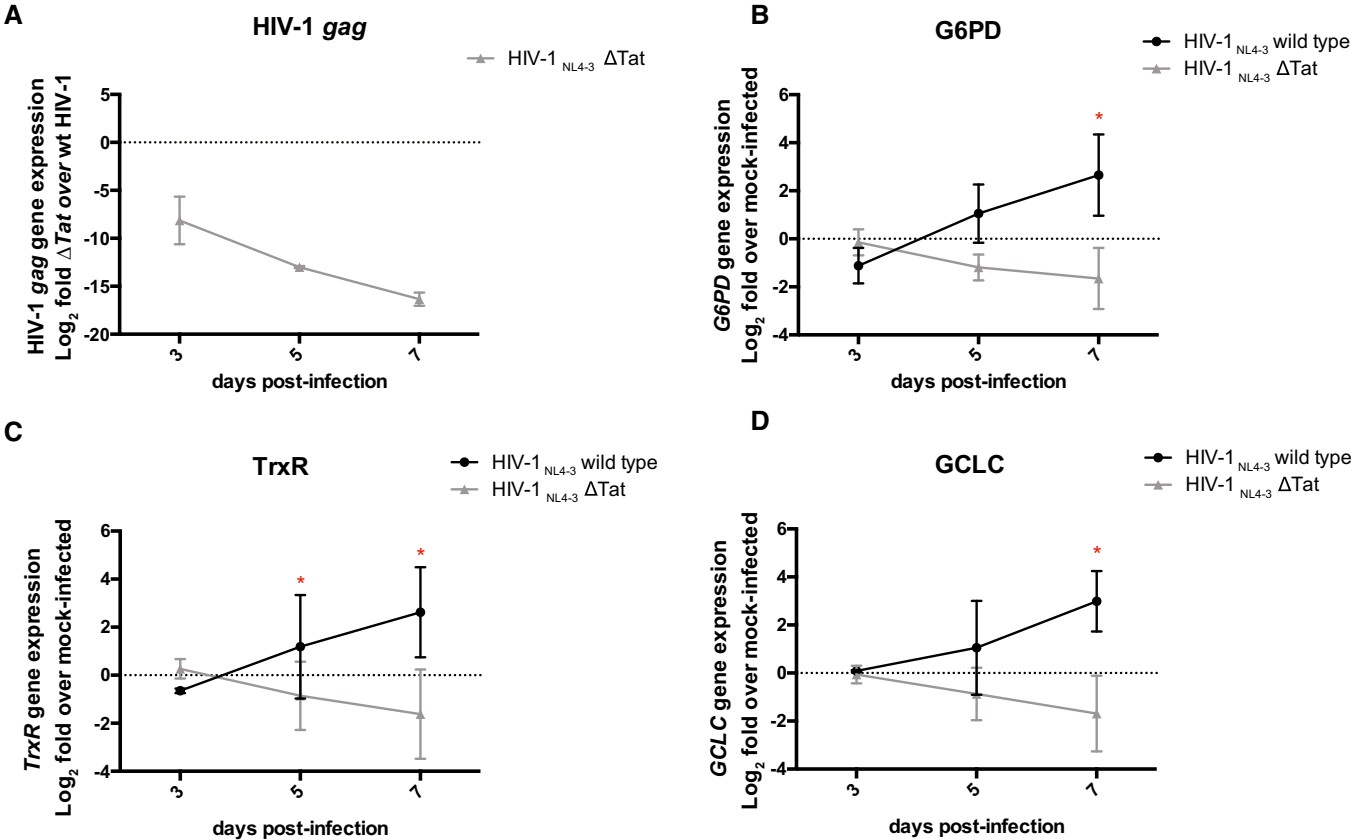

**Figure 4. Relative expression of HIV-1 *gag* and genes regulating the pentose phosphate pathway or antioxidant responses in CD4+ T cells infected with *wild type* and Tat-deficient HIV-1.**

Primary CD4+ T cells were isolated from total blood of healthy donors and activated with α-CD3-CD28 beads for 72 h. Cells were then mock-infected or infected with wild-type HIV-1$_{pNL4-3}$ or with Tat-deficient HIV-1$_{pNL4-3}$ (Bejarano *et al*, 2019). Cells were cultured for 1 week post-infection, and gene expression was measured by qPCR.

A  Relative expression of HIV-1 *gag* in cells infected with Tat-deficient HIV-1$_{pNL4-3}$ as compared to cells infected with wild-type HIV-1 $_{pNL4-3}$.

B–D  Relative expression of the limiting rate enzyme of the pentose phosphate pathway, i.e., G6PD (B), and of genes regulating the thioredoxin and glutathione antioxidant pathways, i.e., TrxR1 (C) and GCLC (D), in HIV-1 infected as compared to mock-infected cells.

Data information: Data were first normalized using 18S as housekeeping control and then expressed as Log$_2$ fold mRNA expression in wild type vs Tat-deficient infection (panel A) or in infected vs mock-infected cells (panels B–D), which were calculated using the 2-ΔΔCT method (Livak & Schmittgen, 2001). GCLC = glutamate—cysteine ligase; TrxR1= thioredoxin reductase 1. Data are expressed as mean ± SD of two replicates and were analyzed by two-way ANOVA followed by Sidak's post-test. *$P < 0.05$.

As G6PD was previously shown to be upregulated in HIV-1-infected primary CD4+ T cells, during the late stages of the infection (Shytaj *et al*, 2020), we decided to further test the role of proviral transcription on the expression of G6PD in a longer time course of infection with a wild type and a previously described Tat-deficient HIV-1 (Bejarano *et al*, 2019). The results showed that infection with Tat-deficient HIV-1 led to lower retroviral transcription (Fig 4A), as expected, and, in contrast to the wild-type HIV-1, was not associated with increased expression of G6PD (Fig 4B). Thus, Tat is required to induce the upregulation of this enzyme during the productive infection stage, and, subsequently, high enzyme levels are then maintained upon latency establishment (Shytaj *et al*, 2020). Of note, this result is in line with the role of the pentose phosphate cycle in fueling pyrimidine synthesis through ribose 5-phosphate (Fig EV4). In line with previous evidence showing that G6PD expression is accompanied by an antioxidant response (Shytaj *et al*, 2020) and with the pro-oxidant activity of Tat (El-Amine *et al*, 2018), the Tat

mutant HIV-1 also failed to induce two main antioxidant genes acting downstream of NADPH production by the pentose phosphate cycle (Louboutin & Strayer, 2014) (Fig 4C and D). In particular, Tat was necessary to induce the expression of thioredoxin reductase 1 (TrxR1) (Fig 4C) and glutamate-cysteine ligase (GCLC) (Fig 4D), which are essential for the function of the thioredoxin (Trx) and glutathione (GSH) pathways, acting downstream of NADPH production.

## Downstream inhibition of the glycolysis-alternative pentose phosphate pathway can induce a "shock and kill" effect in latently infected cells

We then tested whether blocking the downstream, NADPH-mediated antioxidant effects of the pentose phosphate cycle could induce HIV-1 escape from latency and reduce the ability of infected cells to survive oxidative stress.

For this purpose, we chose the drugs auranofin (AF) and buthionine sulfoximine (BSO), which, by inhibiting Trx and GSH regeneration, respectively (Benhar *et al*, 2016), can block the antioxidant/pro-latency effect of NADPH. These drugs were also preferred because of their translational potential, due to their clinical (separately) and pre-clinical (combined) testing as anti-reservoir compounds in PLWH and macaques infected with the simian immunodeficiency virus (SIV) (Benhar *et al*, 2016; Diaz *et al*, 2019). As expected, at the concentrations chosen for the reactivation experiments, the two drugs were able to synergistically increase oxidative stress in a previously described reporter model which allows measuring GSH potential in live cells (Bhaskar *et al*, 2015) (Appendix Fig S6). When we analyzed HIV-1 production in a number of proviral latency models, we found that AF/BSO favored proviral reactivation, at different efficiencies, in both lymphoid and myeloid models (Fig EV5A).

Moreover, when cell viability was analyzed, results showed that combined inhibition of Trx and GSH led to the preferential killing of HIV-1-infected cells as compared to their uninfected counterparts (Fig EV5B), although specific leukemia/lymphoma cell lines were highly sensitive to AF and BSO treatment irrespective of HIV-1 infection, in line with the previously described sensitivity of such neoplasias to these drugs (Fiskus *et al*, 2014; Benhar *et al*, 2016).

To increase the translational relevance of our findings, we tested the combination of AF and BSO in primary-like/primary lymphoid cells (Fig 5), choosing Th17 cells and cells isolated from total blood of PLWH to model, respectively, tissue (Hunt, 2010) and peripheral reservoirs. In line with our results on other cellular models, combined AF/BSO treatment was able to induce HIV-1 reactivation and preferential death of infected Th17 cells (Fig 5A and B). As for cells from PLWH, we initially conducted an analysis of the metabolic profiles of peripheral blood mononuclear cells (PBMCs), showing significant enrichment of the pentose cycle upon treatment with both AF and BSO as compared to cells treated with BSO only (Appendix Fig S7). These effects were not detectable when cells treated with AF-only were compared with cells treated with BSO, thus supporting a specific impact of the drug combination on the pentose cycle. These results are also in line with the well-known compensation of GSH inhibition by the Trx system (Benhar *et al*, 2016), thus confirming that only the combination of both drugs can lead to a sustained pro-oxidant effect. When we tested the therapeutic potential of AF and BSO on $CD4^+$ T cells of PLWH with stably suppressed viremia under ART (Appendix Table S2), it was not possible to separately analyze the viability of the latently infected cells, because of the low frequency of these cells in peripheral blood. We thus chose to measure the effect of the treatment on the selective elimination of HIV-1-infected cells. For this purpose, $CD4^+$ T cells were left untreated or treated for 48 h with AF, BSO, or a combination of the two, and cells were then sorted for viability (Appendix Fig S8A) to eliminate the potential bias of the general cytotoxic effect of AF on lymphocytes (Chirullo *et al*, 2013) (Appendix Fig S8B). The frequency of the total integrated proviral DNA was then measured by Alu-PCR, and normalization of the proviral DNA copies to the number of cells examined allowed to standardize the analysis. The results showed that proviral HIV-1 DNA was significantly lower in the cell cultures that had received both AF and BSO (Fig 5C). Of note, cells from two of the donors showed loss of integrated proviral DNA signal after AF/BSO treatment. In line with the hypothesis of an increased mortality of HIV-1 infected cells carrying intact proviruses, AF and BSO exerted only minor effects on early products of HIV-1 reactivation as measured by *Tat/rev* Induced Limiting Dilution Assay [TILDA (Procopio *et al*, 2015)], which is based on proviral expression in intact cells (Appendix Fig S9). Interestingly, the specificity of the result of proviral DNA decrease is further corroborated by the analysis of general cell culture viability before sorting. Indeed, unlike proviral DNA decrease, the decrease in general cell viability was mainly affected by AF (Appendix S8B), as previously shown (Chirullo *et al*, 2013), and not specifically by the AF/BSO combination.

Overall, these data show that glycolysis downregulation and increased reliance on the pentose cycle in latently HIV-1-infected cells can be exploited by pro-oxidant drugs targeting Trx and GSH to induce HIV-1 reactivation and/or mortality of infected cells.

## Discussion

The present study highlights the central role of downregulation of glycolytic activity as a determinant of the transition from productive to latent HIV-1 infection, an effect that is reversed upon proviral reactivation. By integrating our data with the extant literature, we may now distinguish two phases characterizing the cellular interplay between glycolysis and HIV-1 infection.

In the first phase, HIV-1 preferentially replicates in fully activated cells, a cellular milieu characterized by elevated glycolysis, that is optimal to sustain high levels of viral replication. In line with an initial enhancement of glycolysis upon HIV-1 infection described by others (Valle-Casuso *et al*, 2019; Guo *et al*, 2021), our data show that the HIV-1 transactivator Tat can mediate *Glut1* upregulation, thus favoring the intracellular import of glucose needed by glycolysis, as a mechanism of maximal exploitation of the metabolic machinery of the host cell by the retrovirus.

The second phase, accompanied by changes in host cell metabolism and glycolytic regulation during establishment of proviral

**Figure 5. "Shock and kill" effect of combined thioredoxin (Trx) and glutathione (GSH) inhibition in primary lymphoid models of HIV-1 infection.**

A, B    Reactivation from HIV-1 latency (A) and relative cell viability (B) in primary Th17 cells following 24 h treatment with the Trx inhibitor auranofin (AF; 500 nM), the GSH inhibitor buthionine sulfoximine (BSO; 250 μM), or a combination of the two. GFP-HIV-1 expression was determined by FACS. Data are expressed as mean ± SD of three replicates and were analyzed by one-way ANOVA followed by Tukey's post-test (A) or two-way ANOVA followed by Sidak´s post-test (B). Solid lines represent the means.

C    Levels of integrated HIV-1 DNA following treatment for 48 h with AF and/or BSO in $CD4^+$ T cells derived from PLWH under suppressive antiretroviral therapy. Live cells were sorted after treatment, and integrated DNA was measured by Alu-PCR. The latency reactivating agent SAHA was used as a reference compound (Archin *et al*, 2012). Data were analyzed by non-parametric Friedman´s test followed by Dunn's post-test.

Data information: *$P < 0.05$, **$P < 0.01$, ***$P < 0.001$, ****$P < 0.0001$.

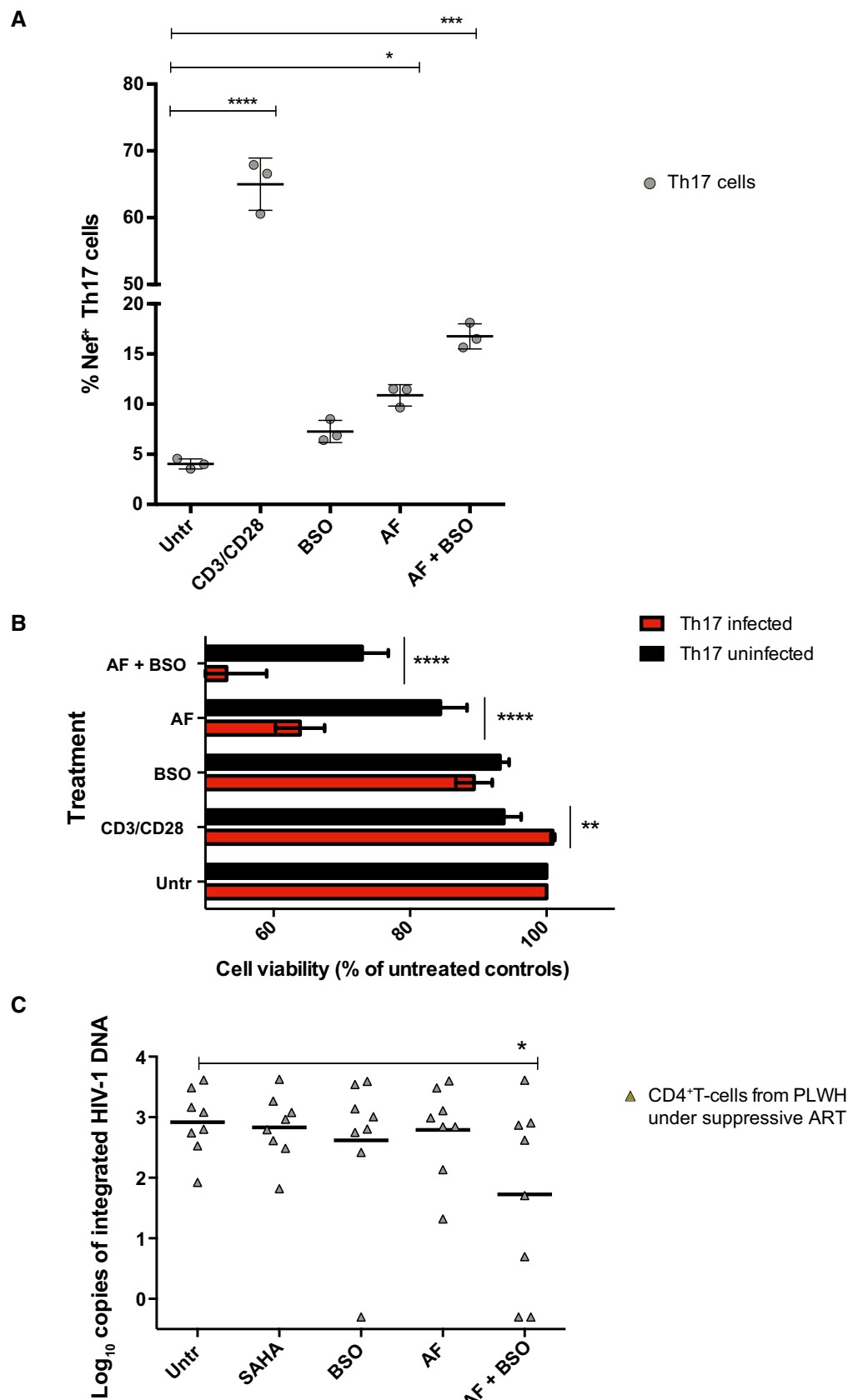

Figure 5.

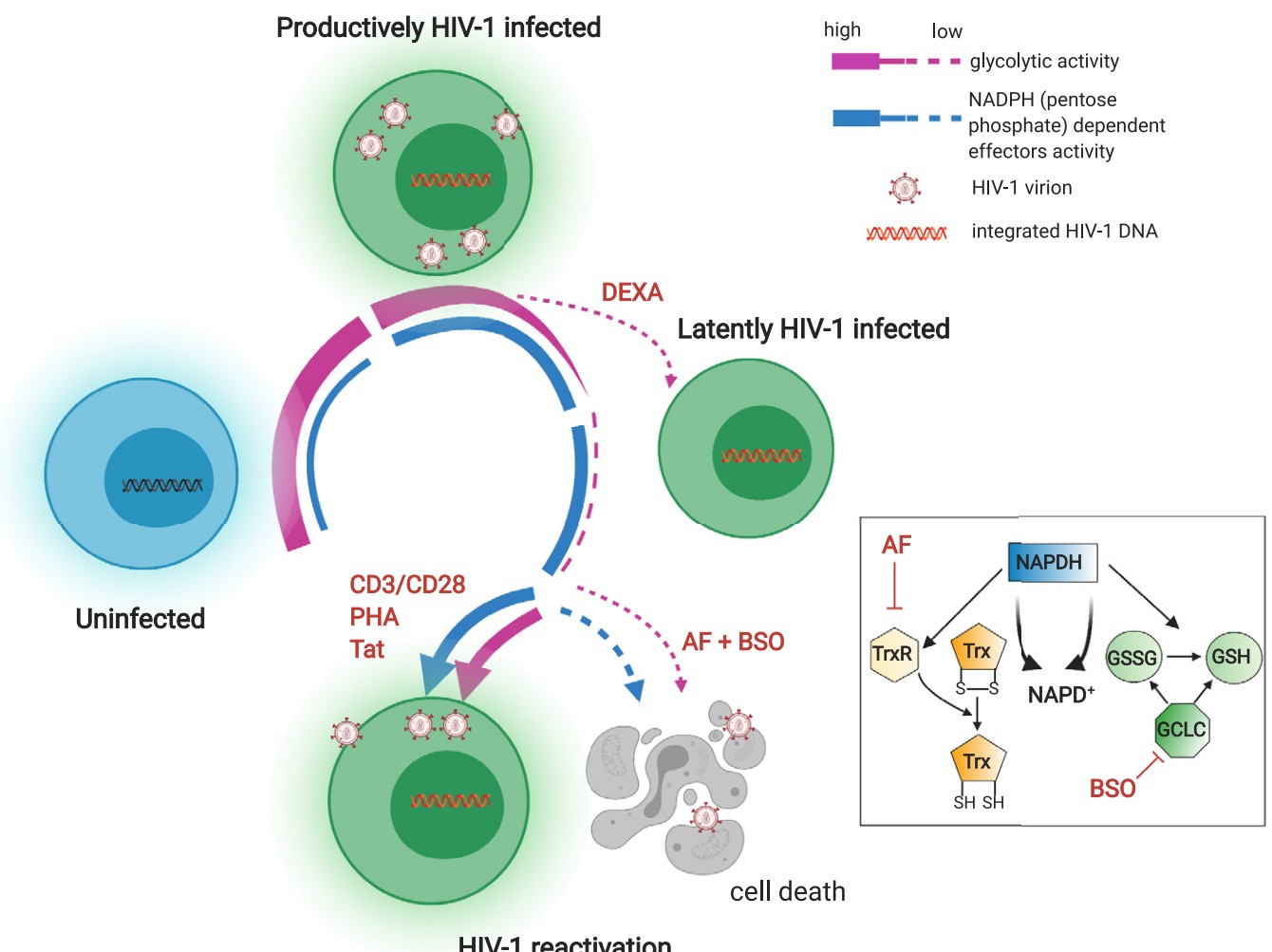

**Figure 6. Glycolysis downregulation is a hallmark of HIV-1 latency and sensitizes infected cells to oxidative stress.**

Schematic model of the regulation of glycolysis during sequential stages of HIV-1 infection and of its potential therapeutic targeting (created with BioRender). Uninfected cells become susceptible to HIV-1 infection upon activation by upregulating a number of metabolic pathways including glycolysis (Valle-Casuso et al, 2019). During productive HIV-1 infection, glycolysis is progressively downregulated, a process which is in part compensated by reliance on the parallel pentose phosphate pathway. Latent HIV-1 infection can ensue either spontaneously, leading to further glycolysis downregulation, or be induced by agents such as dexamethasone (DEXA), which can directly inhibit glycolysis. During latency, the pentose phosphate pathway remains active and fuels, through NADPH activity, regeneration of the two main antioxidant molecules, i.e., Trx and GSH. Reversion to a productively infected state is accompanied by partial reactivation of the glycolytic pathway. Blocking Trx and GSH inhibits downstream effects of NADPH without restoring glycolytic activity, thus favoring death of latently infected cells. TrxR = thioredoxin reductase; GCLC = glutamate-cysteine ligase; GSSG = oxidized GSH.

latency, has not been previously characterized. We show that entry into the latent phase of the infection is accompanied by a profound metabolic rearrangement, with general depression of the energy-producing pathways, in particular glycolysis, and reliance on other energy sources (Fig 6). One such source is the pentose cycle, which provides infected cells with antioxidants that are necessary for HIV-1 latency induction (Shytaj et al, 2020). In this regard, we show upregulation of the interconnected pyrimidine metabolism, which is involved in reactive oxygen species (ROS) production (Tabata et al, 2018). The effects of pyrimidine metabolism might underlie G6PD upregulation, which is observed during late-productive and latent HIV-1 infection (Shytaj et al, 2020). Of note, G6PD can in turn decrease the levels of glucose-6-phosphate, thus potentially explaining downmodulation, by substrate subtraction, of the early

glycolytic enzyme GPI (Dyson & Noltmann, 1968). While our data indicate that some of the metabolic changes are a direct effect of HIV-1, our results are not incompatible with a role of lentivirus-independent factors, such as the reversal to cellular quiescence, that may drive the cell to both glycolysis downregulation and establishment of HIV-1 latency.

The importance of downregulated glycolysis for latency maintenance is further shown by the effect of latency-reversing stimuli, which we show to partially restore glycolytic activity. Again, this may be due to specific retroviral factors such as HIV-1 Tat, but also be an epiphenomenon of general cellular activation. In this regard, single-cell analysis shows that the inability to upregulate glycolysis is associated with impaired viral reactivation, potentially suggesting a link between cellular metabolism and deep latency.

Our analysis of the glycolytic pathway at a single-cell resolution helps to reconcile our results with the works of Palmer et al, which showed that glycolysis is upregulated in PLWH, independently of ART (Palmer et al, 2014a, 2014b). PLWH are characterized by chronic immune hyperactivation (Deeks et al, 2004), also when viral loads are stably suppressed by ART. As the percentage of latently infected cells in vivo is low, it is conceivable that their lower glycolytic activity is masked in vivo by hyperactivated cells and their bystander effects. Another result apparently contradictory to ours was obtained by Castellano et al (2019), who analyzed a time course of macrophages infected in vitro with HIV-1 and concluded that glycolysis might not be impaired in latently infected macrophages. However, their conclusions were based on the analysis of the acidification capability (a byproduct of late glycolysis), while our results, instead, highlight the downmodulation of the early phases of glycolysis in latently infected cells. Indeed, our data are consistent with those of Castellano et al in showing that lactate, the main metabolite responsible for extracellular acidification, as well as its precursor, pyruvate, may even be increased in some latently infected cell types due to the simultaneous activation of other pathways, bypassing the early glycolytic pathway.

Although we focused primarily on glucose metabolism, we were also able to evaluate a number of alternative energy-producing pathways that are modulated in latently infected cells. While a full reconstruction of other pathways will require stepwise analysis in future works, our metabolomic analysis suggests that the Krebs cycle (Wu et al, 2015; Viña et al, 2016), which intervenes downstream of glycolysis, is another biochemical pathway possibly affected by HIV-1 latency. In lymphoid cells, acetyl-CoA, and citrate, the initial metabolites of the Krebs cycle are decreased. In agreement with Castellano et al (2019), our report confirms that, in latently infected myeloid cells, glutamate becomes a readily alternative source for the Krebs cycle. Interestingly, another source for the Krebs cycle supply, lipid degradation, which exerts a positive role in PLWH by favoring anti-HIV-1 immune responses (Loucif et al, 2021), is inactive in late cellular HIV-1 infection.

Although inhibition of glycolysis might become an interesting tool to prevent proviral reservoir establishment (Valle-Casuso et al, 2019), this inhibitory approach does not provide a viable strategy for targeting the viral reservoirs, once they are established. For example, the glycolysis inhibitor 2-DG was able to decrease viral replication and kill the infected cells in newly infected cultures where glycolytic metabolism remained high. On the other hand, data derived from the clinic have shown that enhancers, rather than inhibitors, of glycolysis may be beneficial to PLWH (Planas et al, 2021).

Another hallmark of HIV-1 infection identified by our metabolomic analysis is an altered metabolism of nicotinamide, which is required for the synthesis of $NADP^+$, the precursor of the antioxidant NADPH, which is the main pentose cycle product. Therefore, limited availability of the $NADP^+$/NADPH machinery may contribute to the increased sensitivity of infected cells to inhibitors of the downstream effects of the pentose phosphate cycle. In particular, NADPH acts as a cofactor for the regeneration of the two most important antioxidant defenses, Trx and GSH (Benhar et al, 2016; Miller et al, 2018), implying that dual inhibition of the Trx and GSH pathways might induce a vicious cycle (Fig 6). In this regard, an interesting parallel can be drawn with neoplastic cells, as previous

studies showed increased susceptibility to the cytotoxic effects of combined Trx and GSH pathway inhibition in lung cancer cells with pharmacologically inhibited glycolysis (Fath et al, 2011; Li et al, 2015).

Of particular importance, the drugs used in the present work (AF and BSO) had already shown the potential to decrease the viral reservoir in macaques and PLWH (Shytaj et al, 2015; Diaz et al, 2019). A similar effect was observed in macaques treated with another pro-oxidant compound and Trx pathway inhibitor, i.e., arsenic trioxide (Yang et al, 2019). One characteristic of these strategies was not only the elimination of latently infected cells in vivo but also the enhancement of anti-HIV-1 cell-mediated immunity (Shytaj et al, 2015). In light of the results of the present study, this immune enhancement could be interpreted as a consequence of the "shock" effect against viral latency provided by the combined inhibition of the Trx and GSH pathways. One limitation of the current approach is, however, the lack of complete specificity toward HIV-1-infected cells. Future studies will be required to identify more selective compound combinations or to determine the hormetic dose zone of AF and BSO which be capable of exerting the maximal effects in infected cells, while sparing uninfected cells. Although still controversial for some agents, the concept of a U-shaped dose response due to counteracting responses is becoming widely accepted in redox biology (Zimmermann et al, 2014). However, the present study was bound to well-established drug concentrations able to significantly inhibit their targets.

Taken together, our results highlight glycolysis downregulation as a distinctive metabolic feature of latently HIV-1-infected cells which can potentially be exploited to block entry of viruses into latency or subsequently target latent reservoirs.

# Materials and Methods

### Cell cultures and HIV-1 infection

The following cell lines or primary cells were used as models of productive or latent HIV-1 infection: (i) lymphoid cells: (a) latently HIV-1-infected J-Lat 9.2 cells and uninfected Jurkat T cells, (b) latently HIV-1-infected primary Th17 cells (Alvarez-Carbonell et al, 2017), (c) primary $CD4^+$ T cells from healthy donors infected with HIV-1 in vitro (Shytaj et al, 2020), and (d) primary $CD4^+$ T cells from PLWH with viral loads stably suppressed by ART; (ii) myeloid cells: (a) the THP-1 monocytic cell line, (b) the U937 and U1 (latently HIV-1 infected) promonocytic cell lines, (c) primary human monocyte-derived macrophages (Shytaj et al, 2013), (d) human immortalized microglia (hμglia) clone C20 (uninfected) and clone HC69 (latently infected with HIV-1) (Alvarez-Carbonell et al, 2017; Garcia-Mesa et al, 2017; Llewellyn et al, 2018), and (e) iPSC-derived microglia (iMG) uninfected or infected with HIV-1 (iMG/HIV) (Alvarez-Carbonell et al, 2019); and (iii) reporter TZM-bl cells transfected with HIV-1 Gag-mCherry viruses pseudotyped with JR-FL Env or without Env.

#### Lymphoid cells

The parental Jurkat E6.1 (ATCC number: TIB-152) cell line and the latently HIV-1-infected J-Lat 9.2 cell line (Jordan, 2003) were cultured using RPMI + 10% fetal bovine serum (FBS) and plated at

0.25–0.5 × $10^6$ cells/ml as previously described (Shytaj *et al*, 2020). Latently infected 2D10 Jurkat T cells were cultured as described in Alvarez-Carbonell *et al* (2017).

Latently HIV-1-infected Th17 cells were generated as previously described (Garcia-Mesa *et al*, 2017; Dobrowolski *et al*, 2019). Briefly, naïve CD4$^+$ T cells were isolated using a RoboSep CD4$^+$ Naïve T-cell negative selection kit (STEMCELL Technologies Inc., Vancouver, British Columbia, Canada), and 2 × $10^6$ cells were resuspended in 10 ml RPMI medium and stimulated with 10 µg/ml concanavalin A (ConA) (EMD Millipore, Billerica, MA, USA) in the presence of subset-specific cytokines. Cells were cultured for 72 h at 37°C, followed by addition of 10 ml of fresh medium, additional 10 µg/ml ConA, polarization cocktail cytokines, and 120 IU/ml of IL-2. After 6 days, cells were washed and resuspended in RPMI medium supplemented with the growth cytokines IL-23 (50 ng/ml) and IL-2 (60 IU/ml). Cells were then infected in a 24-well plate using VSV glycoprotein-pseudotyped virus expressing CD8 and GFP at a multiplicity of infection (MOI) of 2 at a cellular concentration of 5 × $10^6$ cells per ml, in the presence of cell subset cytokines. Cells were spinoculated at 2,000 × $g$ for 1.5 h at room temperature and then placed in an incubator overnight. Cells were adjusted to 1 × $10^6$ per ml in the presence of cell subset-appropriate growth cytokines. After 48 h, infection efficiency was determined by GFP expression and infected cells were isolated using RoboSep mouse CD8a-positive selection kit II (STEMCELL Technologies Inc., Vancouver, British Columbia, Canada). Cells (50 × $10^6$ per ml) were pre-incubated with 50 µl/ml of antibody cocktail from the kit and 40 µl/ml of magnetic beads and diluted into 2.5 ml RoboSep buffer. Positive cells were recovered by magnetic bead separation, suspended in 1 ml of medium, and vortexed to release the cells and beads from the tube wall.

Primary CD4$^+$ T cells for *in vitro* HIV-1 infection were isolated from total blood of healthy individuals using the RosetteSep™ Human CD4$^+$ T Cell Enrichment Cocktail (STEMCELL Technologies Inc., Vancouver, British Columbia, Canada) as previously described (Shytaj *et al*, 2020) and according to the manufacturer's instructions. The blood was obtained through the Heidelberg University Hospital Blood Bank following approval by the local ethics committee. To induce activation before HIV-1 infection Dynabeads® Human T-Activator CD3/CD28 was added to cells for 72 h. Cells were then mock-infected or infected using 2 ng p24 of HIV-1$_{pNL4-3}$/$10^6$ cells. Mock-infected and HIV-1 infected cells were then cultured for 2 weeks in RPMI + 20% FBS with 10 ng/ml IL-2 at a density of 0.5–2 × $10^6$/ml. At 3, 7, 9, and 14 days post-infection, 1 × $10^6$ cells were pelleted and used for RNA extraction and transcriptomic analysis (microarray and RNA-Seq) as previously described (Shytaj *et al*, 2020).

For *ex-vivo* experiments on CD4$^+$ T cells or PBMCs of PLWH, cells from adult donors were analyzed. Experiments were conducted in conformity to the principles set out in the WMA Declaration of Helsinki and the Department of Health and Human Services Belmont Report. The use of samples was approved by the Institutional Review Board of the Centre Hospitalier Universitaire Vaudois and by the Human Subjects Review Committee of the Federal University of Sao Paulo. All subjects gave written informed consent. Experiments were exclusively performed on cells isolated from treated PLWH with undetectable viremia (HIV-1 RNA levels < 50 copies per ml of plasma) for at least 12 months. Culture conditions and HIV-1 reactivation experiments were conducted as described in Procopio *et al* (2015).

### Myeloid cells

For producing latently HIV-1-infected THP-1 (ATCC number: TIB-202) cell cultures, uninfected cells were cultured on a 6-well plate at a density of 1 × $10^6$ cells per well in RPMI growth medium containing 10% FBS, 1% penicillin/streptomycin, and 50 nM of 2-mercaptoethanol. Infection with a HIV-1-GFP virus was carried out by spinoculation, as described in Alvarez-Carbonell *et al* (2017). Positively selected cells were placed in RPMI medium with cell type-specific growth cytokines at 1 × $10^6$ in upright flasks and allowed to expand for a week prior to treatments. During all assays, suspension cells were cultured at a density of 1 × $10^6$ cells per ml, in 96-well plates in a volume of 100 µl.

Monocytic cell lines U937 and U1 (harboring the latent HIV-1 provirus) were cultured in RPMI + 10% FBS. Cells were stably transduced to express the Grx1-roGFP2 biosensors in cytoplasm as described previously (Bhaskar *et al*, 2015).

The procedure used to generate hµglia/HIV HC69 from C20 was previously described in Garcia-Mesa *et al* (2017). Cells were cultured in BrainPhysTM medium (STEMCELL Technologies Inc., Vancouver, British Columbia, Canada) containing 1× N2 supplement-A (Gibco, Invitrogen, #17502–048), 1× penicillin–streptomycin (Gibco™, #15140122), 100 µg/ml Normocin™ (InvivoGen, #ant-nr-1), 25 mM glutamine (Gibco ™, #25030081), 1% FBS, and 1 µM DEXA (freshly added to the cell culture) (Sigma-Aldrich, #D4902), as previously described (Garcia-Mesa *et al*, 2017). For experiments, cells were plated at 0.1 × $10^6$ cells per well in 24-well plates.

Human iPSC-derived microglia (Tempo-iMG™; Tempo Bioscience, CA) was cultured in DMEM/F-12 (Thermo Fisher, Waltham, MA, USA) supplemented with 1× N2 supplement, 0.5× NEAA, 2 mM L-Glutamine, 100 ng/ml GM-CSF, 50 ng/ml IL-34, and infected with VSVG-HIV-GFP as previously described (Alvarez-Carbonell *et al*, 2019). The experiments with iMG and iMG/HIV cells (adherent) were performed 72 h post-infection at a density of 50,000 cells per well in a 96-well plate pre-coated with Growth Factor Reduced Matrigel (Corning Inc., Corning, NY, USA).

### TZM-bl cells

TZM-bl (kindly provided by Dr. Quentin Sattentau, University of Oxford) and Lenti-XTM 293T cells (Takara Bio, Clontech, Saint Germain en Laye, France) were cultured in either complete DMEM or complete DMEM F-12 (Thermo Fisher, Waltham, MA, USA), respectively, supplemented with 10% FBS and 1% penicillin–streptomycin. Cells were grown at 37°C in the presence of 5% $CO_2$.

For HIV-1 pseudovirus production, the pR8ΔEnv plasmid (encoding HIV-1 genome harboring a deletion within *env*), pcRev, and NL4-3 Gag-mCherry ΔEnv were kindly provided by Dr. Greg Melikyan (Emory University, Atlanta, GA, USA). The plasmid encoding JR-FL Env was a kind gift from Dr James Binley (Torrey Pines Institute for Molecular Studies, FL, USA). For lactate measurements, Laconic construct was obtained from Addgene (ref. 44238).

HIV-1 pseudovirus bearing Gag-mCherry was produced as described previously (Coomer *et al*, 2020). Briefly, Lenti-XTM 293T cells were transfected at 60–70% confluency with a mix of pR8ΔEnv, pcRev, NL4-3 Gag-mCherry ΔEnv and with or without JR-FL Env, at a 2:1:3:3 ratio, using GeneJuice transfection reagent (Novagen, EMD Millipore, Billerica, MA, USA). 72 h after transfection, viral supernatants were collected, filtered (0.45 µm), and

concentrated using Lenti-XTM Concentrator (Takara Bio, Clontech, Saint Germain en Laye, France).

Tzm-bl cells were plated onto 8-well μ-Slide (Cat. No: 81826, Ibidi, Gräfelfing, Germany) and transfected on the same day with 250 ng of plasmid expressing the Laconic biosensor per well, using GeneJuice transfection reagent. 12 h post-transfection, the medium was replaced with fresh complete DMEM after washing with PBS 1× and cells were incubated for further 4 h at 37°C. 16 h post-transfection, cells were infected at MOI of 1 with HIV-1 Gag-mCherry viruses pseudotyped with JR-FL Env or without Env as a negative control for infection (No Env viruses). Viruses were diluted in a final volume of 100 μl FluoroBrite DMEM 2% FBS (Thermo Fisher, Waltham, MA, USA) per well and added onto cells. Cells were spinoculated at 2,100 g in a refrigerated centrifuge (4°C) for 20 min. The viral inoculum was then removed, and cells were washed with PBS 1× and incubated for further 90 min in FluoroBrite DMEM 2% FBS at 37°C to allow viral fusion to occur. The medium was later replaced by complete DMEM, and cells were again incubated at 37°C. Transfected cells expressing Laconic, challenged to HIV-1 pseudoviruses, were analyzed by Fluorescence Lifetime Microscopy (FLIM) 3 days post-infection as described below.

## RNA extraction

Total cellular RNA was extracted using the InviTrap® Spin Universal RNA Mini Kit (Stratec Biomedical, Germany) according to the manufacturers' instructions and as previously described (Shytaj et al, 2020). RNA concentration was assessed using a P-class P 300 NanoPhotometer (Implen GmbH, Munich, Germany).

## Microarray and RNA-Seq analyses

Primary CD4+ T cells infected with HIV-1 or mock-infected were subjected to microarray and RNA-Seq using 500 ng of total RNA that was quality checked for integrity via Bioanalyzer. Microarray was performed using the HumanHT-12 beadchip (Illumina, Inc., 5200 Illumina Way San Diego, CA 92122 USA) and scanned using an iScan array scanner. Data extraction was done for all beads individually, and outliers were removed when the absolute difference to the median was > 2.5 times MAD (2.5 Hampelís method). Raw data are available at GSE163405.

Bead-level microarray raw data were converted to expression values using the lumi R package for quality control, variance stabilization, normalization, and gene annotation (Du et al, 2008). Briefly, raw data and control probes were loaded in R using the lumiR and addControlData2lumi functions. Raw signals were background corrected, estimating the background based on the control probe information with the bgAdjust method of the lumiB function. Background corrected data were then processed with the variance-stabilizing transformation (VST) of the lumiT function to stabilize the variance and were finally normalized using the quantile normalization implemented in lumiN. Gene expression data were annotated using the R package illuminaHumanv4.db that contains the mappings between Illumina identifiers and gene descriptions.

To identify the impact of HIV-1 infection on gene expression, we compared the expression levels of CD4+ T cells infected with HIV-1 with those of mock-infected cells using Significance Analysis of Microarray (Tusher et al, 2001) algorithm coded in the same R

package. In SAM, we estimated the percentage of false-positive predictions (i.e., false discovery rate, FDR) with 100 permutations and selected as differentially expressed those genes with an FDR q-value ≤ 0.05.

Over-representation analysis was performed using Gene Set Enrichment Analysis (Subramanian et al, 2005) and using the gene sets of the Biocarta and Reactome collections from the Broad Institute Molecular Signatures Database (http://software.broadinstitute.org/gsea/msigdb) as well as a customized gene set derived from AmiGO (see section "Glycolysis pathway" below). GSEA software (http://www.broadinstitute.org/gsea/index.jsp) was applied on $Log_2$ expression data of cells infected with HIV-1 or matched mock-infected controls. Gene sets were considered significantly enriched at FDR < 5% when using Signal2Noise as a metric and 1,000 permutations of gene sets.

Differential expression of genes in RNA-Seq data sets was analyzed using the DESeq2 package (Love et al, 2014). The data sets used were retrieved from Shytaj et al (2020) (GSE127468) for CD4+ T cells and from Garcia-Mesa et al (2017) (SRP075430) for uninfected (C20) and HIV-1-infected (HC69) microglia. Heatmaps were generated using the Morpheus tool (https://software.broadinstitute.org/morpheus).

## ScRNA-Seq analyses

scRNA-Seq data sets were retrieved from Golumbeanu et al (2018) (GSE111727) and from Cohn et al (2018) (GSE104490) and analyzed with the Seurat [version 3.1.5; (Stuart et al, 2019)] R package. In particular, normalized gene expression data for GSE111727 were downloaded from Zenodo repository (Zenodo_Data_S2) and loaded into a Seurat object; raw gene counts for GSE104490 were downloaded from GEO and normalized using Seurat. T-cell subsets were classified using the SingleR R package (Aran et al, 2019). Differentially expressed genes were calculated using the FindMarkers function of the Seurat package.

The scRNA-Seq data of HC69 (GSE163979) were generated as follows: Immortalized human microglia HC69 cells were left untreated or treated with dexamethasone [DEXA, 1 μM (Sigma-Aldrich, St. Louis, MO, USA)] for 72 h. After harvesting, a minimum of 600,000 viable cells per condition was subjected to Drop-Seq as described in Macosko et al (2015). After capturing individual cells in the oil droplets with barcoded beads (ChemGenes, Inc. Wilmington, MA, 01887 USA), droplets were broken, and cDNA libraries were generated using Illumina Nextera XT kit (Illumina, Inc., 5200 Illumina Way San Diego, CA, 92122 USA). Next-generation sequencing and quality control were performed at MedGenome Inc (Foster City, CA, 94404 USA). Drop-Seq Tools v.1.0 and STAR-2.5.1b alignment tools were used to process and map the sequences to the hg18 human reference genome and to the retroviral portion of the HIV-$1_{pNL4-3}$ construct. As a result, we built digital gene expression matrices (DGE) containing the read counts for human and HIV-1 genes. The DGE matrices were used to generate the Seurat objects. After filtering, a total of 6,528 control and 5,869 DEXA-treated individual cell transcriptome profiles were consolidated, each expressing at least 100 genes, with each gene expressed in at least 3 cells. To prevent "zero inflation" bias, further filtering was performed to isolate only cells expressing HIV-1 and the genes of the glycolytic pathway (HUMAN-GLYCOLYSIS). Correlation scatter plots were

built using Seurat, accompanied by the calculations of the Spearman correlation coefficients R and related *P*-values.

## Proteomic analysis

Proteomic analysis of CD4[+] T cells infected *in vitro* with HIV-1 or mock-infected was retrieved from Shytaj *et al* (2020). The data set is available at the ProteomeXchange Consortium via the PRIDE (Perez-Riverol *et al*, 2019) partner repository with the data set identifier PXD012907. Relative protein quantification was performed by Despite Data-independent Acquisition (DIA) processing raw data with Spectronaut Pulsar X (version 11) using default and previously described parameters (Shytaj *et al*, 2020). Heatmaps were generated using the Morpheus tool (https://software.broadinstitute.org/morpheus).

## Metabolomic analysis

Metabolomic analysis was performed as described previously (Li *et al*, 2020). Briefly, polar metabolites were extracted from cells using a cold extraction solution containing 80% methanol and 20% water. Samples were analyzed by LC-MS/MS on a Thermo Q Exactive HF-X mass spectrometer coupled to a Vanquish LC System. LC separation was performed by hydrophilic interaction liquid chromatography (HILIC), pH 9, using a SeQuant ZIC-pHILIC column (MilliporeSigma). Peak areas, representing metabolite levels, were extracted using Thermo Compound Discoverer 3.0. The following data normalization procedures were used for each data set depending on the available parameters: (i) Data derived from microglia cells (HC69 and C20) were normalized to protein amount; (ii) data derived from Jurkat cells (2D10 and E6) were normalized to total intensity of the identified metabolites; and (iii) data derived from PBMCs of PLWH were normalized to equivalent cells injected on column. Metabolites were identified by accurate mass and retention based on pure standards and by accurate mass and MS/MS fragmentation followed by searching the mzCloud database (www.mzcloud.org). For heatmap generation, Log$_2$ fold change values were calculated based on average expression values from triplicate samples. Statistically significant changes were assessed by Student's *t*-test (*P*-value) and the Benjamini–Hochberg false discovery rate (FDR) to account for multiple testing (*q*-value).

Metabolite pathway and enrichment analysis were performed using MetaboAnalyst 4.0 (http://www.metaboanalyst.ca) (Xia *et al*, 2009). For enrichment analysis, a table of raw peak intensity values was submitted to the platform and the Benjamini–Hochberg FDR correction was used to adjust all *P*-values and reduce false-positive discovery for multiple testing. The *q* conversion algorithm was then used to calculate FDRs in multiple comparisons using FDR < 0.05 as threshold for significance in all tests.

For pathway analysis, Human Metabolome Database (HMBD) (Wishart, 2020) IDs of the metabolites differentially expressed between the conditions compared were submitted to the MetaboAnalyst platform. Comparisons were analyzed by Student's *t*-test using Holm's correction for multiple comparisons, so as to generate a *q*-value. The library 'Homo sapiens (human)' of the Human Metabolome Database was used for pathway analysis. For network generation, metabolite data were integrated with the RNA-Seq data of the same microglia model (described in its dedicated methods section). As input data, KEGG IDs, *P*-values, and Log$_2$ fold changes were used

for the selected compounds (metabolites and gene transcripts). In order to analyze a correlation network of the compounds in shared pathways, MetScape (Gao *et al*, 2010), an app implemented in Java and integrated with Cytoscape (version 3.5.1), was used.

## Glycolysis pathway

Except for *a priori* analysis (GSEA and metabolic pathway enrichment), the list of enzymes and metabolites used to define the glycolytic pathway was selected based on known literature and to reflect all metabolic/enzymatic steps associated with glycolysis irrespective of the cell type examined. Specifically, the gene list was retrieved from AmiGO Gene ontology using the filters: GO:0061621 (canonical glycolysis) AND Homo sapiens (pathway named in the paper as: HUMAN-GLYCOLYSIS). The metabolite list was retrieved from the KEGG Pathway database entry M00001 [Glycolysis (Embden-Meyerhof pathway)].

## Drug treatments

For glycolysis inhibition, TZM-bl cells infected with JR-FL without Env (No Env virus) were treated with 100 mM 2-deoxy-glucose (2-DG) (Sigma-Aldrich, St. Louis, MO, USA) 2 h prior to image acquisition.

For viral reactivation and cell viability experiments, cells were treated with auranofin (Sigma-Aldrich #A6733; 500 nM), L-Buthionine-sulfoximine (i.e., Sigma-Aldrich BSO #B2515; 250 μM), or a combination of the two for 24 or 48 h as indicated in the captions of Additional files 8, 9, and 11. In parallel, cells were incubated with one of the following positive control activating agents/-drugs: α-CD3/CD28 beads (1:1 bead-to-cell ratio), TNF (10 ng/ml), 12-O-tetradecanoylphorbol-13-acetate at 10 μM concentration (TPA; Sigma-Aldrich, Saint Louis, MI, USA), and suberoylanilide hydroxamic acid 0.5 μM (SAHA; Selleckchem, Houston, TX, USA S1047).

## Lipid droplets and HIV-1 p24 staining

TZM-bl cells exposed to HIV-1 Gag-mCherry viruses pseudotyped with JR-FL Env (MOI 1) were fixed 3 dpi using 4% paraformaldehyde (PFA). Fixed cells were blocked/permeabilized using 10% fetal bovine serum (FBS), 0.5% saponin in PBS buffer (Immunofluorescence, IF, buffer). Primary antibodies recognizing HIV-1 p24 (ab53841, Abcam) were diluted in IF buffer to 4 μg/ml and applied to cells for 1h in a humid chamber at 37°C. Anti-goat secondary antibodies conjugated to Alexa 488 dye (A-11055, Invitrogen) were diluted to 2 μg/ml in IF buffer and added to cells for 30 min at 37°C. Lipid droplet staining was performed adding Nile Red dye (Cat 60029, Biotium) diluted in PBS to 1 μg/ml, and the preparation was incubated for 10 min at room temperature. Excess of Nile Red dye was removed by washing twice with PBS before microscopy imaging.

## Fluorescence lifetime imaging microscopy (FLIM)

Fluorescence microscopy experiments on fixed TZM-bl cells expressing HIV-1 p24 and stained with Nile Red dye were performed using the PicoQuant MicroTime 200 confocal microscope. Cells were imaged under a 60×/NA 1.20 water objective. HIV-1 p24 expressing cells (immuno-stained with Alexa 488 and Nile Red dye) were

excited using a 485-nm pulsed laser, tuned at 40 MHz, and subsequently detected by a hybrid detector (500–550 nm) and a single-photon avalanche diode (SPAD) detector (650–690 nm). Images were acquired with a laser dwell time of 1.3 µs with a pixel size of 180 nm and obtained after 20 times frame repetitions.

Time-domain FLIM experiments on live TZM-bl cells expressing Laconic were performed using a Time-correlated Single Photon Counting (TCSPC) approach operated by the FALCON module (Leica Microsystems, Manheim, Germany) integrated within the Leica SP8-X-SMD microscope. Cells of interest were selected under a 63×/NA 1.20 water objective. Laconic-expressing cells were excited using a 440-nm pulsed laser, tuned at 40 MHz, and subsequently detected by a hybrid internal detector (475–510 nm) in photon counting mode. Transfected cells co-expressing Laconic and Gag-mCherry were also excited using a DPSS 561 continuous laser, and mCherry fluorescence emission was detected by a hybrid internal detector (525–560 nm). Images were acquired using a scan speed of 200 Hz with a pixel size of 120 nm and obtained after 10 times frame repetitions.

## Image analysis

Fluorescence intensity analysis of confocal microscopy images was performed using ImageJ software. HIV-1-infected cells (p24-Alexa 488 positive) vs uninfected cells (p24-Alexa 488 negative) were identified in the 500–550 nm emission channel. Number of lipid droplets per cell was quantified from the 650–690 nm emission channel using the "Find maxima" tool in ImageJ. Results were statistically analyzed performing an unpaired $t$-test (GraphPad Prism 9.1.0).

FLIM images were analyzed using the FALCON module (Leica Microsystems, Manheim, Germany) integrated within the Leica SP8-X-SMD microscope. Images were binned $3 \times 3$ to reach at least 100 counts/pixel. Individual cells expressing Laconic alone (No Env infected) or co-expressing Gag-mCherry were selected as regions of interest. A two-exponential decay deconvoluted with the Instrument Response Function (IRF) and fitted by a Marquardt nonlinear least-square algorithm was applied to the photon counting histogram with the long lifetime component fixed to 2.6 ns. The average life-time, intensity weighted (TauInt) was calculated per cell and normalized to the mean TauInt of the no-Env condition of each experiment. Normalized results from three independent experiments were statistically analyzed using a one-way ANOVA test (OriginLab software, Northampton, USA).

## MTT assay of cell viability

Cell viability upon treatment with auranofin and/or BSO was measured using the CellTiter 96®Non-Radioactive Cell Proliferation Assay (MTT) (Promega; Madison, WI, USA) according to the manufacturer's instructions, as described in Shytaj et al (2020). Absorbance values were measured using an Infinite 200 PRO (Tecan, Männedorf, Switzerland) plate reader. After blank subtraction, absorbance values were normalized using matched untreated controls.

## Flow cytometry and cell sorting

To measure GFP expression in J-Lat 9.2 cells, $500 \times 10^5$ cells were fixed with 4% PFA in PBS, washed twice with PBS, and resuspended in the FACS buffer. GFP fluorescence was measured using a BD FACSCelesta (Becton Dickinson, Franklin Lakes, NJ, USA) flow cytometer and analyzed using the FlowJo software (FlowJo LLC, Ashland, Oregon, USA v7.6.5).

GFP expression in hμglia/HIV HC69 and HIV-1-infected iPSC-derived microglia was measured using a LSRFortessa instrument for cell sorting, the FACSDiva software (Becton Dickinson, Franklin Lakes, NJ, USA) for data collection, and the WinList 3D software (Verity Software House, Topsham, ME, USA) for data analysis.

Viability of U937 and U1 cells was assessed by propidium iodide staining. Briefly, cells were suspended in PBS and stained with 1.5 µM propidium iodide (PI) for 15 min in the dark. After washing twice with PBS, cells were analyzed on a flow cytometer using the phycoerythrin detector with 488 nm excitation and 575/26 nm emission on a FACSVerse Flow cytometer (Becton Dickinson, Franklin Lakes, NJ, USA).

The expression of CD4 in cells of PLWH was detected with an anti-human-CD4 antibody conjugated to PE-CF594 (BD PharMingen Catalog No. 562281). Cells were sorted based on viability using the PE Annexin V Apoptosis Detection Kit I (BD PharMingen Catalog No. 559763) using a LSR Aria cell sorter and the FACSDiva software (Becton Dickinson, Franklin Lakes, NJ, USA) for data collection.

## Redox potential measurement

Intracellular redox potential measurements in U1 cells were done as described earlier (Bhaskar et al, 2015). Briefly, the ratio-metric response of cells expressing the Grx1-roGFP2 sensor was obtained by measuring excitation at 405 and 488 nm at a fixed emission (510/10 nm) using a FACS Verse Flow cytometer (Becton Dickinson, Franklin Lakes, NJ, USA).

## Real-time PCR and ALU-HIV PCR

The expression of HIV-1 (Gag-p24) in U1 cells and of metabolic genes in U1 and U937 cells were measured by qPCR as described previously (Bhaskar et al, 2015). Briefly, total cellular RNA was reverse transcribed to cDNA (iScriptTM cDNA synthesis kit, Bio-Rad, Hercules, CA, United States). Real-time PCR (iQTM SYBR Green Supermix, Bio-Rad Hercules, CA, United States) was performed using the Bio-Rad C1000TM real-time PCR system. The primers for Gag-24 and housekeeper β-actin genes were described elsewhere (Bhaskar et al, 2015). The following primers were used to amplify metabolic genes: (i) GLUT1, forward CTGCTCATCAACCG CAAC, reverse CTTCTTCTCCCGCATCATCT; (ii) G6PD, forward CTGTTCCGTGAGGACCAGATCT, reverse TGAAGGTGAGGATAACG CAGGC; (iii) CAD, forward GTTTGCAGTCCTTCCCGC, reverse CCGGTTTGAAACACCACTTCC; and (iv) GFPT1, forward CCAGCCA GTTTGTATCCCTT, reverse CAAGCATGATCTCTTTGCGT. Relative fold change levels were calculated using the delta delta CT method. All experiments were done at least twice in triplicate.

The expression of antioxidant genes in primary CD4[+] T cells was measured by qPCR using the conditions and primers described in Shytaj et al (2020).

To measure integrated HIV-1 DNA in live CD4[+] T cells of PLWH, Alu-HIV PCR was performed as described in Chomont et al (2009). HIV-1 reactivation in the same cell types was measured by Tat/rev Induced Limiting Dilution Assay as described in Procopio et al (2015).

**The paper explained**

**Problem**

The HIV epidemic remains a global challenge since current therapies are able to manage, but not cure, the infection. HIV eradication is hampered by the presence of a small subset of infected cells which can harbor the virus in a latent form and survive undetected by antiretroviral therapies or the immune system.

**Results**

We show that latently infected cells are characterized by a specific set of metabolic signatures and, in particular, by decreased activity of glycolysis, which is a pathway otherwise aimed at generating energy by breaking down glucose molecules. Our finding is supported by the convergence of genetic and metabolic data from multiple cell types selected from those known to be the main hiding spots of latent HIV. To further prove the specific connection between glycolysis and HIV latency, we have shown that this metabolic pathway is activated when latent HIV is "awakened" by drugs acting as latency reactivating agents. By proving the reduced reliance on glycolysis of latently HIV-infected cells, our work paves the way to therapies exploiting this metabolic signature. We show that latently infected cells need alternative sources to meet their energy requirements, and in particular the "Pentose phosphate pathway", which generates an energetic metabolite named "NADPH". This metabolite can also play a key detoxifying activity in cells, and we show that drugs blocking this activity can lead to reactivation of latent HIV and death of infected cells.

**Impact**

Finding specific markers of HIV latency can help to characterize the mechanisms used by the virus to establish and maintain its reservoirs. Exploiting these defining metabolic features of latently infected cells can pave the way to new therapies aimed at the selective eradication of infected cells from the body.

**Statistical analysis**

Statistical analysis of *in-silico* data is described in the respective Methods subchapters. HIV-1 reactivation and cell viability data were analyzed by parametric (i.e., one- or two-way ANOVA tests) or non-parametric (i.e., Friedman test). Parametric testing was adopted when normality could be hypothesized (sample size ≤ 3) or restored through an appropriate transformation. Data sets characterized by sample size > 3 which did not pass the normality tests (D'Agostino & Pearson or Shapiro–Wilk) and for which a transformation was not applicable were analyzed by non-parametric tests. For both parametric and non-parametric tests, post-test comparisons were used to compare specific groups as described in the Figure captions. Analyses were performed using GraphPad Prism (GraphPad Software, San Diego, CA, USA).

# Data availability

The data sets produced in this study are available in the following databases:

- Microarray: Gene Expression Omnibus GSE163405 (https://www.ncbi.nlm.nih.gov/geo/query/acc.cgi?acc = GSE163405)
- RNA-Seq data: DDBJ Sequence Read Archive SRP075430 (https://trace.ddbj.nig.ac.jp/DRASearch/study?acc = SRP075430)
- scRNA-Seq: Gene Expression Omnibus GSE163979 (https://www.ncbi.nlm.nih.gov/geo/query/acc.cgi?acc = GSE163979)

**Expanded View** for this article is available online.

## Acknowledgements

The authors thank Dr. Hans Georg Kräusslich and Dr. Thorsten Müller for kindly providing the HIV-1$_{NL4-3}$ Δtat vector. R.S.D. acknowledges support from the Fundação de Amparo à Pesquisa do Estado de São Paulo and the Conselho Nacional de Desenvolvimento Científico e Tecnológico (FAPESP 2013/11323-5; CNPq - 454700-2014-8; CNPq/DECIT 441817/2018-1). I.L.S. acknowledges support from the Humboldt Foundation (Ref 3.3-ITA-1193954-HFST-P) and the Fundação de Amparo à Pesquisa do Estado de São Paulo (Ref. 19/17461-7). A. Si. acknowledges support from the Department of Biotechnology, Indian Institute of Science (# 22-0905-0006-05-987-436). S.P-P and I.C-A work has been supported by the European Research Council (ERC-2019-CoG-863869 FUSION to S.P-P.). The authors thank the Microarray Unit of the Genomics and Proteomics Core Facility, German Cancer Research Center (DKFZ), for providing Expression Profiling services.

## Author contributions

ILS, DA-C, and ASa conceived the project. ILS, FAP, LCN, SP-P, ASi, ML. RSD, JK, DA-C, and ASa designed the experiments. ILS, FAP, ICA, MHM, VKP, SS, NC, BL, and DAC performed *in vitro* experiments. ILS, FAP, IC-A, DA-C, and ASa analyzed *in vitro* data. MT, MF, KL, FY, and SB analyzed transcriptomic data. H-YT and ARG ran the samples for metabolomic profiling and MT analyzed metabolomic data. ILS and ASa wrote the manuscript.

## Conflict of interest

A. Sa is the inventor of a patent covering the use of auranofin and buthionine sulfoximine for the treatment of HIV/AIDS. The other authors declare no conflict of interests.

## For more information

i https://www.natap.org (English).
ii poz.com (English).
iii https://www.treatmentactiongroup.org/cure/trials/ (English).
iv www.aidsmap.com (English).
v www.natap.org (English).
vi https://www.hiv-symptome.de/forum/forums/leben-mit-hiv.14/ (German).
vii https://hivforum.info/forum/index.php (Italian).

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
