## [Review Process File · EMBO Molecular Medicine]

Glycolysis downregulation is a hallmark of HIV-1 latency and sensitizes infected cells to oxidative stress

Iart Luca Shytaj, Francesco Procopio, Mohammad Tarek, Irene Carlon-Andres, Hsin-Yao Tang, Aaron Goldman, MohamedHusen Munshi, Virender Kumar Pal, Mattia Forcato, Sheetal Sreeram, Konstantin Leskov, Fengchun Ye, Bojana Lucic, Nicolly Cruz, Lishomwa Ndhlovu, Silvio Bicciato, Sergi Padilla-Parra, Ricardo Diaz, Amit Singh, Marina Lusic, Jonathan Karn, David Alvarez-Carbonell, and Andrea Savarino

DOI: 10.15252/emmm.202013901

Corresponding authors: Andrea Savarino (andrea.savarino@iss.it) , Iart Luca Shytaj (shytaj.luca@unifesp.br)

Review Timeline:

Submission Date:	3rd Jan 21
Editorial Decision:	16th Feb 21
Revision Received:	16th May 21
Editorial Decision:	14th Jun 21
Revision Received:	25th Jun 21
Accepted:	28th Jun 21

Editor: Jingyi Hou

Transaction Report:

16th Feb 2021

Dear Dr. Savarino,

Thank you again for submitting your work to EMBO Molecular Medicine. One of the referees who initially accepted to review the manuscript finally dropped out. We have now heard back from the other two referees who evaluated your manuscript. As you will see from the reports below, the referees acknowledge the potential interest of the study. However, they also raise substantial concerns about your work, which should be convincingly addressed in a major revision of the present manuscript.

The referees' recommendations are rather clear and there is no need to reiterate their comments. Importantly, direct analysis of cell viability (infected vs. control) needs to be performed, the impact of latent and productive HIV-1 infections on the identified metabolic pathways needs to be assessed. Further, attention should be paid to the discrepancies with previous studies, and limitations with regard to the non-specific effect of metabolic intervention need to be discussed.

During our pre-decision cross-commenting process (in which the referees are given a chance to make additional comments, including on each other's reports), Referee #3 added "limitations with regard to the non-specific effect of metabolic intervention need to be discussed is the main concern for the conclusion and the relevance for this manuscript." Referee #2 said "I totally agree with all the requests and concerns raised by Referee #3. I think that all the issues raised by both referees have to be addressed by the authors before reconsidering the manuscript."

We would welcome the submission of a revised version within three months for further consideration. Please note that EMBO Molecular Medicine strongly supports a single round of revision and that, as acceptance or rejection of the manuscript will depend on another round of review, your responses should be as complete as possible.

We are aware that many laboratories cannot function at full efficiency during the current COVID-19/SARS-CoV-2 pandemic and have therefore extended our "scooping protection policy" to cover the period required for a full revision to address the experimental issues. Please let me know should you need additional time, and also if you see a paper with related content published elsewhere.

I look forward to receiving your revised manuscript.

Sincerely,
Jingyi

Jingyi Hou
Editor
EMBO Molecular Medicine

*** Instructions to submit your revised manuscript ***

** PLEASE NOTE ** As part of the EMBO Publications transparent editorial process initiative (see our Editorial at <https://www.embopress.org/doi/pdf/10.1002/emmm.201000094>), EMBO Molecular Medicine will publish online a Review Process File to accompany accepted manuscripts.

To submit your manuscript, please follow this link:

Link Not Available

- 1) a .docx formatted version of the manuscript text (including Figure legends and tables). Please make sure that the changes are highlighted to be clearly visible to referees and editors alike.
- 2) separate figure files*
- 3) supplemental information as Expanded View and/or Appendix. Please carefully check the authors guidelines for formatting Expanded view and Appendix figures and tables at <https://www.embopress.org/page/journal/17574684/authorguide#expandedview>
- 4) a letter INCLUDING the reviewers' reports and your detailed responses to their comments (as Word file)

Also, and to save some time should your paper be accepted, please read below for additional information regarding some features of our research articles:

- 5) The paper explained: EMBO Molecular Medicine articles are accompanied by a summary of the articles to emphasize the major findings in the paper and their medical implications for the non-specialist reader. Please provide a draft summary of your article highlighting
 - the medical issue you are addressing,
 - the results obtained and

- their clinical impact.

6) For more information: There is space at the end of each article to list relevant web links for further consultation by our readers. Could you identify some relevant ones and provide such information as well? Some examples are patient associations, relevant databases, OMIM/proteins/genes links, author's websites, etc...

7) Author contributions: the contribution of every author must be detailed in a separate section (before the acknowledgments).

8) EMBO Molecular Medicine now requires a complete author checklist (<https://www.embopress.org/page/journal/17574684/authorguide>) to be submitted with all revised manuscripts. Please use the checklist as a guideline for the sort of information we need WITHIN the manuscript as well as in the checklist. This is particularly important for animal reporting, antibody dilutions (missing) and exact p-values and n that should be indicated instead of a range.

9) Every published paper now includes a 'Synopsis' to further enhance discoverability. Synopses are displayed on the journal webpage and are freely accessible to all readers. They include a short stand first (maximum of 300 characters, including space) as well as 2-5 one sentence bullet points that summarise the paper. Please write the bullet points to summarise the key NEW findings. They should be designed to be complementary to the abstract - i.e. not repeat the same text. We encourage inclusion of key acronyms and quantitative information (maximum of 30 words / bullet point). Please use the passive voice. Please attach these in a separate file or send them by email, we will incorporate them accordingly.

You are also welcome to suggest a striking image or visual abstract to illustrate your article. If you do please provide a jpeg file 550 px-wide x 400-px high.

10) A Conflict of Interest statement should be provided in the main text

11) Please note that we now mandate that all corresponding authors list an ORCID digital identifier. This takes <90 seconds to complete. We encourage all authors to supply an ORCID identifier, which will be linked to their name for unambiguous name identification.

Currently, our records indicate that there is no ORCID associated with your account.

Please click the link below to provide an ORCID:

Link Not Available

12) The system will prompt you to fill in your funding and payment information. This will allow Wiley to send you a quote for the article processing charge (APC) in case of acceptance. This quote takes into account any reduction or fee waivers that you may be eligible for. Authors do not need to pay any fees before their manuscript is accepted and transferred to our publisher.

Photos 400-800 DPI

*Additional important information regarding figures and illustrations can be found at <https://bit.ly/EMBOPressFigurePreparationGuideline>

***** Reviewer's comments *****

Referee #2 (Comments on Novelty/Model System for Author):

N/A

Referee #2 (Remarks for Author):

Glycolysis downregulation is a hallmark of HIV-1 latency and sensitizes infected cells to oxidative stress (EMM-2020-13901)

The work presented by Iart Luca Shytaj et al., investigated the regulation of glycolytic metabolism during HIV-1 infection. The authors report that transition to latent HIV-1 infection downregulates glycolysis while viral reactivation inverts this effect. These observations are associated with downregulation of NAD⁺/NADH relaying on NADPH antioxidant maintaining HIV-1 latency. However, blocking NADPH downstream effectors favors HIV-1 reactivation from latently infected myeloid and lymphoid cell lines. This article brings new insights into glucose metabolism/redox system regulation during HIV-1 latency and might suggest new therapeutic strategies for HIV functional cure and reservoir eradication. However, discrepancies with previous studies should be further studied and discussed.

Major comments

1. Previous studies cited by the authors indicate that glycolysis and glucose transport are increased in HIV-1 infected cells. Moreover, two studies reported that glycolysis inhibition induces the death of infected CD4⁺ T cells and that HIV-1 latently infected macrophages metabolically carry no changes in glycolysis (Castellano P et al., Scientific Reports 2019, Valle-Casuso JC et al., Cell Met 2019). Thus, it's difficult to link these previous results to the conclusion obtained by the authors in the first part of their Results section (downregulation of the expression of glycolytic enzymes in HIV-1 productively and latently infected CD4⁺ T-cells as well as the absence of HC69 cell death following treatment with glycolysis inhibitor).

2. It is difficult to distinguish the experiments carried out by the authors and the results exploited to generate the RNA-seq or single-cell RNA-seq data presented in Figure 1. The experiments conducted in Figure 3 to test the ability of two drugs (auranofin and buthionine sulfoximine) to induce viral reactivation show the results obtained in cells from different origins (primary cells from

PLWH and uninfected monocytes derived macrophages or lymphoid and myeloid cells lines) in the same graph. This kind of presentation is not suitable because it does not allow the reader to evaluate the efficacy of the drug depending on the cell type. In addition, it does not allow to see the results obtained on an appropriate number of experiments or patient cells. Similar comment for the cell viability following auranofin and buthionine sulfoximine treatment in Figure 3 B.

3. In Figure 3 C, the authors indicate that "direct and precise analysis of cell viability (infected vs. control) was not possible due to the low frequency of HIV-1-infected cells". However, these data are critical and have to be added to the manuscript since Figure 3 B shows an important effect on cell viability of the combination treatment auranofin+buthionine sulfoximine. The decrease in integrated proviral DNA could be linked to cell death. In addition, patients' characteristics from Figure 3C including viremia and transcriptional status have to be reported in the manuscript.

4. Importantly, the authors have to perform functional experiments to assess the impact of latent and productive HIV-1 infections on the most significant enriched pathways they identified such as the antioxidants metabolism or purine/pyrimidine pathways (glutamate metabolism).

Minor comments

1. In Figure 2B, the decrease in lactate level in cells infected with HIV-1JR-FL 3dpi is difficult to observe.

2. In Figure 2 F, glucose metabolism in productively-infected cells does not seem to be less impaired compared to latently-infected cells (Figure 2 E).

Referee #3 (Comments on Novelty/Model System for Author):

Glycolysis downregulation is a hallmark of HIV-1 latency and sensitizes infected cells to oxidative stress

Luca Shytaj et al.

Neuro-immunometabolism is the new frontier in HIV cure research. It is well established that CD4 T cells are preferentially infected when glycolytic pathway is activated when cell respond to an immune response or in the context of non-specific inflammation, often assessed by the expression of the glucose transporter GLUT-1.

The glycolysis/redox state during HIV-1 infection influence autophagy that regulates cell fate. Investigators used elegant technology with transcriptomic, proteomic, and metabolomic datasets, with single cell analyses to show as expected that less energy is required for latency evidenced by the downplay of glycolysis and re-increases after activation.

Such slowdown of glycolysis is accompanied with a higher reliance on the antioxidant thioredoxin (Trx) and glutathione (GSH) systems linked to stemness. Results point to exploit cell specific glycolytic effect to reduce HIV reservoir and mitochondria to be the great regulator. However, Hormesis could make stress a more complex issue as little stress might be better than no stress or too much stress limiting some experiment interpretation.

Zimmermann A, et al.. When less is more: hormesis against stress and disease. *Microb Cell*. 2014

Investigations are timely and novel but highly dependent of the model used and have Lymphoid and myeloid analyses make the reading of the text and context difficult and long.

In the introduction other metabolic pathway are also under investigation for HIV cure and should be shortly discussed with new reference.

A significant limitation of the study is that it focused only on glycolysis as the energy-producing pathway, while lipophagy is also at play and contribute to naïve and memory status, memory T cells in HIV infection. Loucif H, et al. . Lipophagy confers a key metabolic advantage that ensures protective CD8 T-cell responses against HIV-1. *Autophagy*. 2021 Jan 18:1-16.

Cd4 T cell population is a heterogeneous group with main Th 1, Th17, Th Follicular and Treg functions, which show plasticity and metabolism preference, and been regulated by mitochondrial activity. Loucif H, Plasticity in T-cell mitochondrial metabolism: A necessary peacekeeper during the troubled times of persistent HIV-1 infection. *Cytokine Growth Factor Rev*. 2020.

Major limitation of the metabolic intervention is the non-specific effect, not targeting HIV infected cells but any cells. Arsenic trioxide already used in clinic for acute myeloid leukemia type 3 are relatively toxic and act on a specific type representing only 3 of AML. For HIV the low difference of latent infected CD4 cells vs. long-lived Th17 with stemness will not be very different on a metabolism basis. However, strategies propose based on study findings if can be specific for a cell subtype may affect elimination of latently infected cells while may boost of anti-HIV-1 cell mediated immunity, as reported by Loucif et al. 2020.

However, several issues have to be addressed:

A possible approach to reach this goal is the investigation of the metabolic pathways exploited by the retrovirus to actively replicate and enter a latent state. This sentence is too vague, no clear that replication of virus (virions) lead to latent state, some infection in cell programmed to become long term memory with some stemness may drive the latency for the virus and not the virus inducing latency.

This individual had been treated with a combination of intensified ART and the NAD⁺ precursor nicotinamide, thus suggesting a possible contribution of glycolysis regulation in the therapeutic result obtained. Ref should be added Lebouché B, et al. . Impact of extended-release niacin on immune activation in HIV-infected immunological non-responders on effective antiretroviral therapy. *HIV Res Clin Pract*. 2021 Jan 6:1-9.

Data presented are complex and combine immunology with biochemistry the focus should be on CD4 T cells, not sure that myeloid cell analyses and their heterogeneity and tissue dependence may be better described in another paper.

Educative table should be added with glycolysis as the energy-producing pathway with NADH and the Krebs cycle to better understand study findings and their interactions.

Glycolysis downregulation is a hallmark of HIV-1 latency and sensitizes infected cells to oxidative stress

lart Luca Shytaj et al.

Neuro-immunometabolism is the new frontier in HIV cure research. It is well established that CD4 T cells are preferentially infected when glycolytic pathway is activated when cell respond to an immune response or in the context of non-specific inflammation, often assessed by the expression of the glucose transporter GLUT-1.

The glycolysis/redox state during HIV-1 infection influence autophagy that regulates cell fate. Investigators used elegant technology with transcriptomic, proteomic, and metabolomic datasets, with single cell analyses to show as expected that less energy is required for latency evidenced by

the downplay of glycolysis and re-increases after activation.

Such slowdown of glycolysis is accompanied with a higher reliance on the antioxidant thioredoxin (Trx) and glutathione (GSH) systems linked to stemness. Results point to exploit cell specific glycolytic effect to reduce HIV reservoir and mitochondria to be the great regulator. However, Hormesis could make stress a more complex issue as little stress might be better than no stress or too much stress limiting some experiment interpretation.

Zimmermann A, et al.. When less is more: hormesis against stress and disease. Microb Cell. 2014

Investigations are timely and novel but highly dependent of the model used and have Lymphoid and myeloid analyses make the reading of the text and context difficult and long.

In the introduction other metabolic pathway are also under investigation for HIV cure and should be shortly discussed with new reference.

To limit the study to T cells and remove myeloid cells

To assess naive vs memory cells separated by flow to decipher subset of CD4 T cells according to glycolysis status

clinical impact may be limited as intervention are non specific as apply to any human cells

Referee #3 (Remarks for Author):

Glycolysis downregulation is a hallmark of HIV-1 latency and sensitizes infected cells to oxidative stress

Luca Shytaj et al.

Neuro-immunometabolism is the new frontier in HIV cure research. It is well established that CD4 T cells are preferentially infected when glycolytic pathway is activated when cells respond to an immune response or in the context of non-specific inflammation, often assessed by the expression of the glucose transporter GLUT-1.

The glycolysis/redox state during HIV-1 infection influences autophagy that regulates cell fate.

Investigators used elegant technology with transcriptomic, proteomic, and metabolomic datasets, with single cell analyses to show as expected that less energy is required for latency evidenced by the downplay of glycolysis and re-increases after activation.

Such slowdown of glycolysis is accompanied with a higher reliance on the antioxidant thioredoxin (Trx) and glutathione (GSH) systems linked to stemness. Results point to exploit cell specific glycolytic effect to reduce HIV reservoir and mitochondria to be the great regulator. However, Hormesis could make stress a more complex issue as little stress might be better than no stress or too much stress limiting some experiment interpretation.

Zimmermann A, et al.. When less is more: hormesis against stress and disease. *Microb Cell*. 2014

Investigations are timely and novel but highly dependent of the model used and have Lymphoid and myeloid analyses make the reading of the text and context difficult and long.

In the introduction other metabolic pathway are also under investigation for HIV cure and should be shortly discussed with new reference.

A significant limitation of the study is that it focused only on glycolysis as the energy-producing pathway, while lipophagy is also at play and contribute to naïve and memory status, memory T cells in HIV infection. Loucif H, et al. . Lipophagy confers a key metabolic advantage that ensures protective CD8 T-cell responses against HIV-1. *Autophagy*. 2021 Jan 18:1-16.

Cd4 T cell population is an heterogeneous group with main Th 1, Th17, Th Follicular and Treg functions, which show plasticity and metabolism preference, and been regulated by mitochondrial activity. Loucif H, Plasticity in T-cell mitochondrial metabolism: A necessary peacekeeper during the troubled times of persistent HIV-1 infection. *Cytokine Growth Factor Rev*. 2020.

Major limitation of the metabolic intervention is the non-specific effect, not targeting HIV infected cells but any cells. Arsenic trioxide already used in clinic for acute myeloid leukemia type 3 are relatively toxic and act on a specific type representing only 3 of AML. For HIV the low difference of latent infected CD4 cells vs. long-lived Th17 with stemness will not be very different on a metabolism basis. However, strategies propose based on study findings if can be specific for a cell subtype may affect elimination of latently infected cells while may boost of anti-HIV-1 cell mediated immunity, as reported by Loucif et al. 2020.

However, several issues have to be addressed:

A possible approach to reach this goal is the investigation of the metabolic pathways exploited by the retrovirus to actively replicate and enter a latent state. This sentence is too vague, no clear that replication of virus (virions) lead to latent state, some infection in cell programmed to become long term memory with some stemness may drive the latency for the virus and not the virus inducing latency.

This individual had been treated with a combination of intensified ART and the NAD⁺ precursor nicotinamide, thus suggesting a possible contribution of glycolysis regulation in the therapeutic result obtained. Ref should be added Lebouché B, et al. . Impact of extended-release niacin on immune activation in HIV-infected immunological non-responders on effective antiretroviral therapy. *HIV Res Clin Pract*. 2021 Jan 6:1-9.

Data presented are complex and combine immunology with biochemistry the focus should be on CD4 T cells, not sure that myeloid cell analyses and their heterogeneity and tissue dependence may be better described in another paper.

Educative table should be added with glycolysis as the energy-producing pathway with NADH and the Krebs cycle to better understand study findings and their interactions.

Glycolysis downregulation is a hallmark of HIV-1 latency and sensitizes infected cells to oxidative stress

Luca Shytaj et al.

Neuro-immunometabolism is the new frontier in HIV cure research. It is well established that CD4 T cells are preferentially infected when glycolytic pathway is activated when cells respond to an immune response or in the context of non-specific inflammation, often assessed by the expression of the glucose transporter GLUT-1.

The glycolysis/redox state during HIV-1 infection influences autophagy that regulates cell fate. Investigators used elegant technology with transcriptomic, proteomic, and metabolomic datasets, with single cell analyses to show as expected that less energy is required for latency evidenced by the downplay of glycolysis and re-increases after activation.

Such slowdown of glycolysis is accompanied with a higher reliance on the antioxidant thioredoxin (Trx) and glutathione (GSH) systems linked to stemness. Results point to exploit cell specific glycolytic effect to reduce HIV reservoir and mitochondria to be the great regulator. However, Hormesis could make stress a more complex issue as little stress might be better than no stress or too much stress limiting some experiment interpretation.

Zimmermann A, et al. When less is more: hormesis against stress and disease. *Microb Cell*. 2014

Investigations are timely and novel but highly dependent of the model used and have Lymphoid and myeloid analyses make the reading of the text and context difficult and long.

In the introduction other metabolic pathways are also under investigation for HIV cure and should be shortly discussed with new reference.

A significant limitation of the study is that it focused only on glycolysis as the energy-producing pathway, while lipophagy is also at play and contributes to naïve and memory status, memory T cells in HIV infection. Loucif H, et al. Lipophagy confers a key metabolic advantage that ensures protective CD8 T-cell responses against HIV-1. *Autophagy*. 2021 Jan 18:1-16.

CD4 T cell population is a heterogeneous group with main Th 1, Th17, Th Follicular and Treg functions, which show plasticity and metabolism preference, and been regulated by mitochondrial activity. Loucif H, Plasticity in T-cell mitochondrial metabolism: A necessary peacekeeper during the troubled times of persistent HIV-1 infection. *Cytokine Growth Factor Rev*. 2020.

Major limitation of the metabolic intervention is the non-specific effect, not targeting HIV infected cells but any cells. Arsenic trioxide already used in clinic for acute myeloid leukemia type 3 are relatively toxic and act on a specific type representing only 3% of AML. For HIV the low difference of latent infected CD4 cells vs. long-lived Th17 with stemness will not be very different on a metabolism basis. However, strategies proposed based on study findings if can be specific for a cell subtype may affect elimination of latently infected cells while may boost of anti-HIV-1 cell mediated immunity, as reported by Loucif et al. 2020.

However, several issues have to be addressed:

A possible approach to reach this goal is the investigation of the metabolic pathways exploited by the retrovirus to actively replicate and enter a latent state. This sentence is too vague, no clear that replication of virus (virions) lead to latent state, some infection in cell programmed to become long term memory with some stemness may drive the latency for the virus and not the virus inducing latency.

This individual had been treated with a combination of intensified ART and the NAD⁺ precursor nicotinamide, thus suggesting a possible contribution of glycolysis regulation in the therapeutic result obtained. Ref should be added Lebouché B, et al. . Impact of extended-release niacin on immune activation in HIV-infected immunological non-responders on effective antiretroviral therapy. HIV Res Clin Pract. 2021 Jan 6:1-9.

Data presented are complex and combine immunology with biochemistry the focus should be on CD4 T cells, not sure that myeloid cell analyses and their heterogeneity and tissue dependence may be better described in another paper.

Educative table should be added with glycolysis as the energy-producing pathway with NADH and the Krebs cycle to better understand study findings and their interactions.

Glycolysis downregulation is a hallmark of HIV-1 latency and sensitizes infected cells to oxidative stress
Iart Luca Shytaj et al.

Neuro-immunometabolism is the new frontier in HIV cure research. It is well established that CD4 T cells are preferentially infected when glycolytic pathway is activated when cells respond to an immune response or in the context of non-specific inflammation, often assessed by the expression of the glucose transporter GLUT-1.

The glycolysis/redox state during HIV-1 infection influences autophagy that regulates cell fate. Investigators used elegant technology with transcriptomic, proteomic, and metabolomic datasets, with single cell analyses to show as expected that less energy is required for latency evidenced by the downplay of glycolysis and re-increases after activation.

Such slowdown of glycolysis is accompanied with a higher reliance on the antioxidant thioredoxin (Trx) and glutathione (GSH) systems linked to stemness. Results point to exploit cell specific glycolytic effect to reduce HIV reservoir and mitochondria to be the great regulator. However, Hormesis could make stress a more complex issue as little stress might be better than no stress or too much stress limiting some experiment interpretation.

Zimmermann A, et al.. When less is more: hormesis against stress and disease. Microb Cell. 2014

Investigations are timely and novel but highly dependent of the model used and have Lymphoid and myeloid analyses make the reading of the text and context difficult and long.

In the introduction other metabolic pathways are also under investigation for HIV cure and should be shortly discussed with new reference.

A significant limitation of the study is that it focused only on glycolysis as the energy-producing pathway, while lipophagy is also at play and contributes to naïve and memory status, memory T cells in HIV infection. Loucif H, et al. . Lipophagy confers a key metabolic advantage that ensures protective CD8 T-cell responses against HIV-1. Autophagy. 2021 Jan 18:1-16.

Cd4 T cell population is a heterogeneous group with main Th 1, Th17, Th Follicular and Treg functions, which show plasticity and metabolism preference, and been regulated by mitochondrial

activity. Loucif H, Plasticity in T-cell mitochondrial metabolism: A necessary peacekeeper during the troubled times of persistent HIV-1 infection. Cytokine Growth Factor Rev. 2020.

Major limitation of the metabolic intervention is the non-specific effect, not targeting HIV infected cells but any cells. Arsenic trioxide already used in clinic for acute myeloid leukemia type 3 are relatively toxic and act on a specific type representing only 3 of AML. For HIV the low difference of latent infected CD4 cells vs. long-lived Th17 with stemness will not be very different on a metabolism basis. However, strategies propose based on study findings if can be specific for a cell subtype may affect elimination of latently infected cells while may boost of anti-HIV-1 cell mediated immunity, as reported by Loucif et al. 2020.

However, several issues have to be addressed:

A possible approach to reach this goal is the investigation of the metabolic pathways exploited by the retrovirus to actively replicate and enter a latent state. This sentence is too vague, no clear that replication of virus (virions) lead to latent state, some infection in cell programmed to become long term memory with some stemness may drive the latency for the virus and not the virus inducing latency.

This individual had been treated with a combination of intensified ART and the NAD⁺ precursor nicotinamide, thus suggesting a possible contribution of glycolysis regulation in the therapeutic result obtained. Ref should be added Lebouché B, et al. . Impact of extended-release niacin on immune activation in HIV-infected immunological non-responders on effective antiretroviral therapy. HIV Res Clin Pract. 2021 Jan 6:1-9.

Data presented are complex and combine immunology with biochemistry the focus should be on CD4 T cells, not sure that myeloid cell analyses and their heterogeneity and tissue dependence may be better described in another paper.

Educative table should be added with glycolysis as the energy-producing pathway with NADH and the Krebs cycle to better understand study findings and their interactions.

Glycolysis downregulation is a hallmark of HIV-1 latency and sensitizes infected cells to oxidative stress

lart Luca Shytaj et al.

Neuro-immunometabolism is the new frontier in HIV cure research. It is well established that CD4 T cells are preferentially infected when glycolytic pathway is activated when cell respond to an immune response or in the context of non-specific inflammation, often assessed by the expression of the glucose transporter GLUT-1.

The glycolysis/redox state during HIV-1 infection influence autophagy that regulates cell fate. Investigators used elegant technology with transcriptomic, proteomic, and metabolomic datasets, with single cell analyses to show as expected that less energy is required for latency evidenced by the downplay of glycolysis and re-increases after activation.

Such slowdown of glycolysis is accompanied with a higher reliance on the antioxidant thioredoxin (Trx) and glutathione (GSH) systems linked to stemness. Results point to exploit cell specific glycolytic effect to reduce HIV reservoir and mitochondria to be the great regulator. However, Hormesis could make stress a more complex issue as little stress might be better than no stress or too much stress limiting some experiment interpretation.

Zimmermann A, et al.. When less is more: hormesis against stress and disease. *Microb Cell*. 2014

Investigations are timely and novel but highly dependent of the model used and have Lymphoid and myeloid analyses make the reading of the text and context difficult and long.

In the introduction other metabolic pathway are also under investigation for HIV cure and should be shortly discussed with new reference.

A significant limitation of the study is that it focused only on glycolysis as the energy-producing pathway, while lipophagy is also at play and contribute to naïve and memory status, memory T cells in HIV infection. Loucif H, et al. . Lipophagy confers a key metabolic advantage that ensures protective CD8 T-cell responses against HIV-1. *Autophagy*. 2021 Jan 18:1-16.

Cd4 T cell population is an heterogeneous group with main Th 1, Th17, Th Follicular and Treg functions, which show plasticity and metabolism preference, and been regulated by mitochondrial activity. Loucif H, Plasticity in T-cell mitochondrial metabolism: A necessary peacekeeper during the troubled times of persistent HIV-1 infection. *Cytokine Growth Factor Rev*. 2020.

Major limitation of the metabolic intervention is the non-specific effect, not targeting HIV infected cells but any cells. Arsenic trioxide already used in clinic for acute myeloid leukemia type 3 are relatively toxic and act on a specific type representing only 3 of AML. For HIV the low difference of latent infected CD4 cells vs. long-lived Th17 with stemness will not be very different on a metabolism basis. However, strategies propose based on study findings if can be specific for a cell subtype may affect elimination of latently infected cells while may boost of anti-HIV-1 cell mediated immunity, as reported by Loucif et al. 2020.

However, several issues have to be addressed:

A possible approach to reach this goal is the investigation of the metabolic pathways exploited by the retrovirus to actively replicate and enter a latent state. This sentence is too vague, no clear that replication of virus (virions) lead to latent state, some infection in cell programmed to become long term memory with some stemness may drive the latency for the virus and not the virus inducing latency.

This individual had been treated with a combination of intensified ART and the NAD⁺ precursor nicotinamide, thus suggesting a possible contribution of glycolysis regulation in the therapeutic result obtained. Ref should be added Lebouché B, et al. . Impact of extended-release niacin on immune activation in HIV-infected immunological non-responders on effective antiretroviral therapy. *HIV Res Clin Pract*. 2021 Jan 6:1-9.

Data presented are complex and combine immunology with biochemistry the focus should be on CD4 T cells, not sure that myeloid cell analyses and their heterogeneity and tissue dependence may be better described in another paper.

Educative table should be added with glycolysis as the energy-producing pathway with NADH and the Krebs cycle to better understand study findings and their interactions.

Please find our point-by-point response in red below each comment. To facilitate matching our comments to the new results, we have attached to this reply a copy of the revised figures/panels that are most important to support our answers.

General summary on the Referees' comments:

The referees' recommendations are rather clear and there is no need to reiterate their comments. Importantly, direct analysis of cell viability (infected vs. control) needs to be performed, the impact of latent and productive HIV-1 infections on the identified metabolic pathways needs to be assessed. Further, attention should be paid to the discrepancies with previous studies, and limitations with regard to the non-specific effect of metabolic intervention need to be discussed.

During our pre-decision cross-commenting process (in which the referees are given a chance to make additional comments, including on each other's reports), Referee #3 added "limitations with regard to the non-specific effect of metabolic intervention need to be discussed is the main concern for the conclusion and the relevance for this manuscript." Referee #2 said "I totally agree with all the requests and concerns raised by Referee #3. I think that all the issues raised by both referees have to be addressed by the authors before reconsidering the manuscript."

Regarding the direct analysis of cell viability, we now provide the data requested (Appendix 10B of the revised manuscript). These data are derived from the same experimental set up from which the data on proviral DNA were generated and demonstrate the general viability of the cells, *i.e.* that of the entire CD4⁺ T-cell culture comprising the patients' uninfected cells plus a small minority of infected cells. The whole cultures respond differently to auranofin and buthionine sulfoximine (BSO) in terms of cell viability than in terms of proviral DNA. Starting from the standpoint that we don't want to claim that auranofin has no impact on cell viability [we published a paper on this topic (Chirullo et al. Cell Death Disease 2013)], the data on the whole cell cultures show that the mortality is mainly driven by the previously described effect of auranofin. There is no significant difference when the viability of the cultures treated with auranofin and the auranofin/BSO combination are compared. Instead, the effect of auranofin alone is minimal and non-significant on proviral DNA, and it is the addition of BSO to auranofin to be pivotal for HIV-1 DNA elimination. We believe that this is a strong argument in favor of a specific impact of the auranofin/BSO combination on proviral DNA.

Moreover, we would like to clarify one point: cells from people living with HIV (PLWH) in which the proviral DNA was shown to be decreased by our strategy based on auranofin and BSO were sorted for cell viability before measuring integrated viral DNA. This measurement was normalized on the basis of the number of cells, calculated using the housekeeper gene CD3. We apologize for not being clearer in the former version of the manuscript on this topic. We have now included a figure detailing the sorting strategy used (Appendix 10A of the revised manuscript). In addition, the rationale of the experiment has now been more thoroughly described in the Results section. As mentioned in the previous paragraph, in patient-derived cells, latently infected cells are rare and mixed in a much larger population of surrounding uninfected cells. Therefore, while we did assess the viability of the whole cell population, it was not possible to specifically isolate the viability of those HIV-1 reservoir cells that harbor the virus from that of the surrounding cells of the same individual. For this reason, our strategy of combining cell sorting for viability and measurement of integrated HIV-1 DNA (normalized to the number of cells analyzed) is, in our view, the best possible setup to prove the preferential targeting of latently HIV-1 infected cells by auranofin and BSO in *ex vivo* samples of PLWH. On the other hand, in cellular models in which uninfected and infected cells could be clearly separated, we specifically compared cell viability in the presence or absence of auranofin/BSO, thus providing a further assessment of the preferential effect of the drug combination on latently infected cells (Figure 5B and Figure EV5 in the revised manuscript).

We therefore believe that what we propose is a step forward in the specific targeting of the viral reservoirs, although some off-target effects of course still remain. This limitation has been addressed in the new version of the Discussion.

As for the role of HIV-1 as a driver of metabolic changes, we now provide new data supporting the specific impact of the infection on the metabolic pathways identified. In two new sets of data, as

detailed below, we now show that the HIV-1 accessory protein Tat, which is essential for viral transcription, determines, in the short term, an upregulation in markers of glycolysis and pyrimidine metabolism, which are necessary for retroviral replication. In the longer term, we show that Tat is fundamental for the upregulation of a key gene of the pentose phosphate cycle and the related antioxidant response (as requested by Referee 2). As for latent infection, we provide a new dataset, based on a homogeneously infected lymphoid cell line, showing that the latently infected cells maintain features acquired gradually during the late stages of productive infection and that some of these features are lost upon reactivation from latency. As these metabolic signatures are not present in uninfected cells or cells infected with Tat-deficient HIV-1 or in the same latently infected cell population after it subjected to reactivation from latency, we show that the late stages of HIV-1 replication induce key metabolic features favoring latency establishment. Of note, the use of a lymphoid cell line in the revised version of the manuscript, which is latently infected, but not quiescent, and is proliferating, contributes to exclude that the key metabolic changes observed in lymphocyte cultures undergoing HIV-1 latency are due only to reversal to a quiescent state. However, to complement our findings and acknowledge the possible contribution of cellular quiescence *per se*, we have now addressed this aspect specifically in the Discussion.

Referee 2:

The work presented by Iart Luca Shytaj et al., investigated the regulation of glycolytic metabolism during HIV-1 infection. The authors report that transition to latent HIV-1 infection downregulates glycolysis while viral reactivation inverts this effect. These observations are associated with downregulation of NAD⁺/NADH relaying on NADPH antioxidant maintaining HIV-1 latency. However, blocking NADPH downstream effectors favors HIV-1 reactivation from latently infected myeloid and lymphoid cell lines. This article brings new insights into glucose metabolism/redox system regulation during HIV-1 latency and might suggest new therapeutic strategies for HIV functional cure and reservoir eradication. However, discrepancies with previous studies should be further studied and discussed.

Major comments

1. Previous studies cited by the authors indicate that glycolysis and glucose transport are increased in HIV-1 infected cells. Moreover, two studies reported that glycolysis inhibition induces the death of infected CD4⁺ T cells and that HIV-1 latently infected macrophages metabolically carry no changes in glycolysis (Castellano P et al., Scientific Reports 2019, Valle-Casuso JC et al., Cell Met 2019). Thus, it's difficult to link these previous results to the conclusion obtained by the authors in the first part of their Results section (downregulation of the expression of glycolytic enzymes in HIV-1 productively and latently infected CD4⁺ T-cells as well as the absence of HC69 cell death following treatment with glycolysis inhibitor).

We agree that the original version of the manuscript left some ambiguities when comparing our work to the previous literature. We believe that the revised version of the manuscript fully clarifies these seeming discrepancies and reconciles our results with the major findings of other groups. At the same time, we have tried to better explain and expand our new findings in the manuscript, which regards latency and latency reactivation.

In particular, we have now stressed the main novelty of our work, *i.e.* the fact that we have focused our attention to the late stages of the infection, especially the development of latent HIV-1 infection. The work of Valle Casuso *et al.* (Cell Metabolism 2020) highlighted a higher susceptibility to glycolysis inhibition upon HIV-1 infection, also showing that infection *per se* was preferentially observed in cells characterized by a more active glycolytic metabolism. Our findings do not contradict these observations, but, instead, expand the analysis to encompass the later stages of infection. Indeed, in accordance with Valle Casuso *et al.* our results show that active glycolysis is absolutely necessary for optimally productive HIV infection (as shown by the glycolysis upregulation that we observe upon latency reactivation). Since our data prove that glycolysis is gradually downregulated during transition to HIV latency, it is not surprising that, once latency is established, glycolysis inhibition becomes insufficient to kill the infected cells. In line with this, our data show that latently infected HC69 cells do not succumb when treated with the glycolysis inhibitor dexamethasone. Decreasing glycolysis during latency is expected to lead to a deep latent state (in agreement with our analysis of the scRNA-Seq data of Golumbeanu et al. Cell Reports). Instead, Valle Casuso *et al.* show the preferential killing of recently infected cells by the glycolysis inhibitor 2DG, in line with the increased requirement of glycolysis early upon infection and, as shown by our data, upon latency reactivation.

Also regarding the data published by Castellano *et al.* (2019), we now show that our conclusions do not contradict their findings. In particular, our results, now confirmed by metabolomic analysis of another lymphoid cell model (Figure 3 of the Revised the manuscript), show that the early steps of glycolysis, which are exclusively characteristic of this pathway, are always and specifically affected by HIV-1 latency. On the other hand, late glycolytic metabolites, which are shared with the intertwined pentose phosphate pathway, can be replenished by the latter, depending on the cell type considered. Indeed, Castellano *et al.* based their conclusion on the production of lactate, which is a late glycolytic product. In line with their finding, our revised results show that lactate can be increased in specific cell types upon latency, and this concept is now shown in Figure 3, Figure EV4 and Appendix S4 in the revised manuscript, and also clearly stated in the revised Discussion.

2. It is difficult to distinguish the experiments carried out by the authors and the results exploited to

generate the RNA-seq or single-cell RNA-seq data presented in Figure 1. The experiments conducted in Figure 3 to test the ability of two drugs (auranofin and buthionine sulfoximine) to induce viral reactivation show the results obtained in cells from different origins (primary cells from PLWH and uninfected monocytes derived macrophages or lymphoid and myeloid cells lines) in the same graph. This kind of presentation is not suitable because it does not allow the reader to evaluate the efficacy of the drug depending on the cell type. In addition, it does not allow to see the results obtained on an appropriate number of experiments or patient cells. Similar comment for the cell viability following auranofin and buthionine sulfoximine treatment in Figure 3 B.

We have clarified the representation of the results in Figure 1 and Figure 3 of the revised manuscript. Specifically: the microarray and RNA-Seq analyses on primary CD4⁺ T-cells use our own model to capture the infection transitioning from productive to latent HIV-1 infection (the features of this model were extensively validated in our previous paper: Shtaj et al. EMBO Journal 2020). The results of figure 1D, E were obtained reanalyzing datasets accessed from Golumbeanu et al. (Cell Reports 2018) which was based on a different primary CD4⁺ T-cell latency model. The results of Figure 1E were derived from the analysis of data of Cohn et al. (Nature Med. 2018) using sorted cells of PLWH under suppressive ART. Finally, the results of previous Figure 1G,H (which have been moved to Figure EV2 in the revised manuscript) are based on our own sc-RNA-Seq analysis of HC69 cells. We believe that combining datasets generated in our labs with datasets generated for other purposes in independent laboratories can increase the reproducibility of our observations. However, according to the suggestion, the origin of each dataset is now clarified in detail in the revised version of the manuscript. Moreover, according to this Referee's suggestion as well as to similar suggestions from Referee 3, previous Figure 3, now Figure 5 in the revised manuscript, only shows data derived from lymphoid cells and Figure EV5 of the revised manuscript shows the latency reactivation and viability effect separately for each cell type considered.

3. In Figure 3 C, the authors indicate that "direct and precise analysis of cell viability (infected vs. control) was not possible due to the low frequency of HIV-1-infected cells". However, these data are critical and have to be added to the manuscript since Figure 3 B shows an important effect on cell viability of the combination treatment auranofin+buthionine sulfoximine. The decrease in integrated proviral DNA could be linked to cell death. In addition, patients' characteristics from Figure 3C including viremia and transcriptional status have to be reported in the manuscript.

Please note the general answer to this question in the section "General summary on the Referees' comments" above. Specifically, we apologize for the unclear description of the results in the previous version of the manuscript. The viability that could not be precisely measured was that of cells harboring latent HIV-1 DNA as compared to the surrounding uninfected cells in the same culture. We initially considered taking into account a FISH technique to address this issue. However, the precise quantification of viral DNA content using techniques based on fluorescence would be imprecise and could be still confounded by the presence of dead cells in the original population. The viability of the patients' whole cell population, instead, was taken into account, and is included in the figure below (Appendix S10B of the revised manuscript). These data are derived from the same experimental set up from which the data on proviral DNA were generated and demonstrate the general viability of the cells, *i.e.* that of the entire CD4⁺ T-cell culture comprising the patients' uninfected cells plus a small minority of infected cells. The whole cultures respond differently to auranofin and buthionine sulfoximine (BSO) in terms of cell viability than in terms of proviral DNA. As mentioned above, the impact of auranofin on lymphocyte viability had already been proven before (Chirullo et al. Cell Death and Disease 2013)]. Accordingly, the data on the whole cell cultures show that the mortality is mainly driven by the previously described effect of auranofin. In line with this, there is no significant difference when the viability of the cultures treated with auranofin and the auranofin/BSO combination are compared (Appendix S10B of the revised manuscript). Instead, the effect of auranofin alone is minimal and non-significant on proviral DNA (Figure 5C of the revised manuscript), and it is the addition of BSO to auranofin to be pivotal for HIV-1 DNA elimination. We believe that this is a strong argument in favor of a specific impact of the auranofin/BSO on proviral DNA.

Moreover, also the possible bias due to cell mortality was addressed by sorting of live cells before performing integrated HIV-1 DNA measurement, as detailed in the reply to the “General summary on the Referees’ comments” above and as shown below and in the (Appendix S10B of the revised manuscript). Finally, the description of the experimental rationale and results have been now rephrased at the end of the Results subchapter “Downstream inhibition of the glycolysis-alternative pentose phosphate pathway can induce a “shock and kill” effect in latently infected cells”

Appendix 10 of the revised manuscript. Gating strategy and viability of CD4⁺ T-cells of PLWH under ART left untreated or treated with auranofin (AF) and/or buthionine sulfoximine (BSO). CD4⁺ T-cells

were treated with auranofin AF (500 nM), BSO (250 μ M) or a combination of the two, for 24 h. Viable CD4⁺ T-cells were sorted according to the gating strategy shown in Panel A. Viability data are shown as mean \pm SD (Panel B). Data were analyzed by repeated measures one-way ANOVA followed by Tukey's post-test ** $p < 0.01$.

4. Importantly, the authors have to perform functional experiments to assess the impact of latent and productive HIV-1 infections on the most significant enriched pathways they identified such as the antioxidant metabolism or purine/pyrimidine pathways (glutamate metabolism).

To address this question, we have added a set of experiments isolating the effect of HIV-1 transcription (modulated through either exogenous administration of the Tat protein or genetic deletion of the *Tat* gene) on the expression of key genes of glycolysis, the pentose phosphate pathway, the antioxidant pathways thioredoxin/glutathione and glutamine metabolism. These newly included results show that:

- 1) In U1 cells, in which the Tat/TAR axis is not functional, incubation with exogenous Tat concomitantly increases expression of HIV-1 (as expected) and of *GLUT-1* (please see below Appendix 6 of the revised manuscript). This result is in line with one main finding of the manuscript, *i.e.* that HIV-1 reactivation is accompanied by glycolysis upregulation. Moreover, in the same experimental setup, Tat induced upregulation of Carbamoyl-Phosphate Synthetase 2 (*CAD*), which is required for pyrimidine synthesis from L-glutamine and aspartate. Interestingly, upregulation of *CAD*, but not of *GLUT-1*, was also observed in uninfected U937 cells, suggesting that Tat *per se* could play a direct role on pyrimidine metabolism. Finally, Tat administration led to an increase, albeit not significant, in the expression of G6PD, which is the enzyme initiating the pentose pathway leading to NADPH production. As we had previously shown that expression of G6PD and of other antioxidant factors is increased during late productive and latent HIV-1 infection (Shytaj et al. EMBO Journal 2020) we further analyzed their modulation in primary CD4⁺ T-cells infected with wild type and Tat-deficient HIV-1 during early and late productive infection.
- 2) Experiments in CD4⁺ T-cells showed that infection with Tat-deficient HIV-1 led to impaired viral transcription (as expected) and, unlike infection with wild type HIV-1, was not associated with upregulation of *G6PD* (please see Figure 4 of the revised manuscript below). We then considered the role of Tat-deficiency on the expression of the antioxidant genes thioredoxin reductase (*TrxR*) and glutamate cysteine ligase (*GCLC*), which encode for the downstream antioxidant targets of NADPH that are inhibited by AF and BSO. Interestingly, infection with Tat-deficient HIV-1 was not associated with an increase in the expression of each of the two genes (panels C,D).

Overall, these new experiments show that increasing HIV-1 expression can *per se* upregulate the glycolytic potential as well as the expression of key genes of the pentose phosphate pathway and of glutamine metabolism. Please also see the new results subchapter “*HIV-1 transcription is a specific driver of metabolic changes*” and our reply to the “General summary on the Referees’ comments” above.

Appendix 6 of the revised manuscript. Viro-metabolic effects of exogenous Tat administration in HIV-1 infected and uninfected cells. The HIV-1 infected (U1, Panels A,B) and uninfected (U937, Panel C) cell lines, both deficient for Tat signaling, were left untreated or treated with HIV subtype-B Tat protein (800 ng/ml) for 72 h. Panel A. HIV-1 reactivation as measured by qPCR of *gag* expression in infected cells. Panels B,C. Expression of key metabolic genes of glycolysis (*GLUT1*), the pentose phosphate pathway (*G6PD*), pyrimidine biosynthesis/glutamate metabolism (*CAD*) and hexosamine biosynthesis/glutamate metabolism (*GFPT1*) as measured by qPCR in both infected (B) and uninfected cells (C). Fold change variations over the untreated control were calculated using the $2^{-\Delta\Delta CT}$ method as in (Livak & Schmittgen, 2001). Results are expressed as mean \pm SD and are representative of data from two independent experiments (N=2). Results were analyzed by unpaired t-test with Welch's correction (A) or two-way ANOVA followed by Bonferroni's multiple comparison test (B,C). * $p < 0.05$; ** $p < 0.01$.

Figure 4. Relative expression of HIV-1 gag and genes regulating the pentose phosphate pathway or antioxidant responses in CD4⁺ T-cells infected with wild type and Tat-deficient HIV-1. Primary CD4⁺ T-cells were isolated from total blood of healthy donors and activated with α -CD3-CD28 beads for 72 h. Cells were then mock-infected or infected with wild type HIV-1_{pNL4-3} or with Tat-deficient HIV-1_{pNL4-3} (Bejarano et al., 2019). Cells were cultured for one week post-infection and gene expression was measured by qPCR. Panel A. Relative expression of HIV-1 *gag* in cells infected with Tat-deficient HIV-1_{pNL4-3} as compared to cells

infected with wild type HIV-1_{pNL4-3}. Panels B-D) relative expression of the limiting rate enzyme of the pentose phosphate pathway, *i.e.* G6PD (B), and of genes regulating the thioredoxin and glutathione antioxidant pathways, *i.e.* TrxR1 (C) and GCLC (D), in HIV-1 infected as compared to mock infected cells. Data were first normalized using 18S as housekeeping control and then expressed as Log₂ fold mRNA expression in wild type vs Tat-deficient infection (panel A) or in infected vs mock infected cells (panels B-D), which were calculated using the 2- $\Delta\Delta$ CT method (Livak & Schmittgen, 2001).
GCLC= Glutamate—cysteine ligase; TrxR1= thioredoxin reductase 1.

Minor comments

1. In Figure 2B, the decrease in lactate level in cells infected with HIV-1JR-FL 3dpi is difficult to observe.

In light of this comment and of our new data showing that lactate, as other late glycolytic products, can be replenished by intertwined pathways in some cell types, this Figure has now been moved to the supplementary information (Appendix S4) and the role of pathways other than glycolysis in determining the levels of late glycolytic products is now addressed in detail in the Results and Discussion of the revised manuscript (in particular please see subchapter “*Decreased initial glycolytic metabolism during latent HIV-1 infection*” of the Results).

2. In Figure 2 F, glucose metabolism in productively-infected cells does not seem to be less impaired compared to latently-infected cells (Figure 2 E).

We agree with the Referee on the lack of clarity in our previous data presentation. The difference is now much more evident after providing a direct comparison between latently infected cells and cells in which HIV-1 was reactivated (Figure 2C, Figure EV3C of the revised manuscript).

However, it needs to be highlighted that the development of latency has emerged as a gradual process. Thus, we agree with this Reviewer that the differences between these two states should not be overinterpreted and we have amended our text accordingly throughout the manuscript.

Referee #3 (Comments on Novelty/Model System for Author):

Neuro-immunometabolism is the new frontier in HIV cure research. It is well established that CD4 T cells are preferentially infected when glycolytic pathway is activated when cell respond to an immune response or in the context of non-specific inflammation, often assessed by the expression of the glucose transporter GLUT-1. The glycolysis/redox state during HIV-1 infection influence autophagy that regulates cell fate. Investigators used elegant technology with transcriptomic, proteomic, and metabolomic datasets, with single cell analyses to show as expected that less energy is required for latency evidenced by the downplay of glycolysis and re-increases after activation. Such slowdown of glycolysis is accompanied with a higher reliance on the antioxidant thioredoxin (Trx) and glutathione (GSH) systems linked to stemness. Results point to exploit cell specific glycolytic effect to reduce HIV reservoir and mitochondria to be the great regulator. However, Hormesis could make stress a more complex issue as little stress might be better than no stress or too much stress limiting some experiment interpretation. Zimmermann A, et al.. When less is more: hormesis against stress and disease. Microb Cell. 2014

We agree that GLUT1 is a crucial factor for glucose influx and glycolysis regulation. In this regard, we have now included an experiment specifically isolating the role of HIV-1 transcription on *GLUT1* expression (please see the reply to question 4 of Reviewer 2 and Appendix Figure 6 in the revised manuscript). We also agree that defining the hormetic dose zone of AF and BSO (or of similar compounds) might be crucial to achieve enough selectivity of the treatment and target with higher preference latently infected cells. While we are currently working in that direction, and we have preliminary data suggesting that significantly lower BSO concentrations might be sufficient to target infected cells, our experiments in the paper are a proof-of-concept and thus are conducted using well established drug concentrations equal to *in vitro* equivalents of standard doses administered in clinical trials or animal experiments with AF and BSO. Moreover, the aim of the present study is not only translational but also targeted at pathway discovery. Therefore, we needed to adhere to the AF and

BSO dosages reported in the literature to inhibit their molecular targets efficiently. In this regard, we have now acknowledged at the end of the revised Discussion the need for future studies to identify the hormetic dose interval in order to achieve optimal selectivity with this strategy and we have accordingly quoted the paper suggested by the Referee.

Investigations are timely and novel but highly dependent of the model used and have Lymphoid and myeloid analyses make the reading of the text and context difficult and long.

In light of this comment and of comment 3 of Referee 2, the main figures of the revised manuscript now only contain data derived from lymphoid cells. In particular, we have included a novel metabolomic data set of lymphoid Jurkat T-cells (please see below Figure 2 of the revised manuscript) which consolidates our finding that glycolytic metabolites are downregulated during latent infection, except those that can be replenished by intertwined pathways (such as the pentose phosphate pathway). All data derived from other cell types have been moved to the Extended View or Appendix Figures. We think it appropriate to keep the other cell types as supplementary information as HIV-1 reservoirs are known to be highly heterogeneous. Therefore, the possibility to compare lymphoid cells with other cell types allows to identify those metabolic pathways that are broadly conserved markers of latency (*i.e.* downregulation of early glycolytic metabolites and reliance on pentose phosphate activity) from those that are cell-type dependent (*e.g.* late glycolytic metabolites).

Figure 2 of the revised manuscript. Modulation of glycolysis and other metabolic pathways during productive or latent HIV-1 infection. Latently HIV-1 infected (2D10) or uninfected (E6) Jurkat T-cells were subjected to metabolomic analysis under unstimulated conditions or following stimulation with TNF to reactivate latent HIV-1. Panels A,B) Metabolite enrichment analysis in latently infected cells as compared to their uninfected counterparts (A) and cells reactivated from latency and compared to latently infected cells (B) Jurkat T-cells cells, as compared to their uninfected counterparts. The top enriched pathways were ordered

according to p values obtained by Q statistics for metabolic datasets performed with Globaltest (MetaboAnalyst) (Xia et al., 2009). Panel C) Heatmaps of glycolytic metabolites in latently infected Jurkat T-cells as compared to their uninfected counterparts or Jurkat T-cells with HIV-1 reactivated by TNF as compared to latently infected cells. Data are displayed as Log₂ fold change expression. Adjusted P values (q values) were calculated by the Benjamini-Hochberg false discovery rate. Panels D,E) Relative ratios of NADH/NAD⁺ (D) and ATP/ADP (E) in latently infected and reactivated cells. Data were normalized using the matching uninfected control.

A significant limitation of the study is that it focused only on glycolysis as the energy-producing pathway, while lipophagy is also at play and contribute to naïve and memory status, memory T cells in HIV infection. Loucif H, et al. . Lipophagy confers a key metabolic advantage that ensures protective CD8 T-cell responses against HIV-1. Autophagy. 2021 Jan 18:1-16.

We wish to thank this Referee for the interesting suggestion. We have expanded our analysis in order to take into account this alternative pathway of energy production. Our metabolomic data, including the new metabolomic analysis performed in Jurkat T-cells (see our reply to the first point raised by this Referee and Figure 3 of the revised manuscript), shows that lipid metabolism is downregulated in latently infected cells, and that this downregulation is in line with the less efficient tricarboxylic acid cycle observed in lymphoid cells. Interestingly, we observed an increase in the number (but not density) of lipid droplets in infected cells (please see the Appendix Figure 5 of the revised manuscript below) consistent with a downregulation of lipid catabolism. We interpret these new data as consistent with a switch off, during latent infection, of those pathways more active in the presence of efficient immunity such as that described by Loucif et al. (2021). We have now discussed this concept in the revised version of the Discussion.

Appendix 5. Single-cell lipid droplet content in HIV-1 infected and uninfected TZM-bl cells. Panel A) Micrograph showing TZM-bl cells expressing HIV-1 p24 3 days post-infection (Alexa 488 positive cells; emission 500 nm -550 nm, left column) and stained with Nile Red dye for lipid droplet detection (emission 650 nm – 690 nm, middle column). Merge of green and red channels is shown on the right column. Dashed lines indicate the cytoplasm contour of each cell. Scale bar = 20µm. Panels B-D) Scatter plots showing the comparative analysis of infected (A488 positive) vs non-infected cells (A488 negative) cells. Each dot represents the value obtained for a single cell. Bars represent the mean and error bars the standard deviation of at least 30 cells per condition. (B) Quantification of the absolute number of lipid droplets per cell. (C) Quantification of the size (area) of individual cells. (D) Quantification of the number of lipid droplets detected per cell normalized by the size of each cell. Results were analysed using unpaired t-test. **** p< 0.0001.

Cd4 T cell population is an heterogeneous group with main Th 1, Th17, Th Follicular and Treg functions, which show plasticity and metabolism preference, and been regulated by mitochondrial activity. Loucif H, Plasticity in T-cell mitochondrial metabolism: A necessary peacekeeper during the troubled times of persistent HIV-1 infection. Cytokine Growth Factor Rev. 2020.

We agree that different T-cell subtypes might contribute variably to the effects that we observed. In our models, TH follicular cells, as well as Th17 are not expected to be widely represented (mainly because our primary cells are derived from peripheral blood rather than from lymph nodes). In order to address the contribution of different CD4⁺ T-cell subsets to our results we:

- 1) Analyzed separately the CD4⁺ T-cell subsets which could be identified in the scRNA-Seq dataset of Golumbeanu *et al.* (Cell Reports 2018). The results showed that cluster 1 (the deep latent cluster) was characterized by lower expression of the glycolytic pathway, irrespective of the subset considered, further supporting downregulated glycolysis as a specific marker of HIV-1 latency (please see the Figure EV1 of the revised manuscript, shown below).
- 2) We have now included the experiment on Th17 cells, showing the effect of the AF/BSO combination, to the main Figures (Figure 5A,B of the revised manuscript)

Of note, since the data are derived from peripheral blood cells that had undergone activation before HIV-1 infection, the proportion of naïve cells in these models is necessarily low. Therefore, we have now included an acknowledgment of the limitation of the study in terms of exclusive focus on peripheral blood total T-cells.

Figure EV1 of the revised manuscript. Single-cell transcriptional modulation of glycolytic enzymes in CD4⁺ T-cell subsets. The scRNA-Seq expression of the entire glycolytic pathway was analyzed in primary CD4⁺ T-cells infected *in vitro* with VSVG-HIV-1-GFP and sorted for viral expression as detailed in (Golumbeanu *et al.*, 2018). Following latency establishment, cells were left untreated or HIV-1 expression was reactivated through suberoyl anilide hydroxamic acid (SAHA) or α -CD3-CD28 engagement. Clusters 1 (low latency reactivation potential) and 2 (high latency reactivation potential) were identified by principal component analysis as described in (Golumbeanu *et al.*, 2018). T cell subtypes were identified using the SingleR R package and allocated to each cluster irrespective of the *in vitro* treatment. The expression level of the HUMAN-GLYCOLYSIS pathway was calculated as the average expression of genes comprising the gene list.

Major limitation of the metabolic intervention is the non-specific effect, not targeting HIV infected cells but any cells. Arsenic trioxide already used in clinic for acute myeloid leukemia type 3 are relatively toxic and act on a specific type representing only 3 of AML.

We agree with this limitation. As mentioned in the reply to question 1 of this Referee our drug treatment in this study was designed as a proof of concept based on dosages previously administered in clinical trials or animal experiments of AF and BSO. However, the selectivity range of these and other drugs with a similar mechanism of action might be improved when targeting specifically the HIV-1 latent reservoir. This point is now mentioned in the revised Discussion. We would also like to

add that, while a better selectivity for latent cells might be achievable, the temporary elimination of part of the uninfected cells, if well tolerated, might be both desirable and necessary to achieve a sustained HIV-1 remission without antiretrovirals (functional cure), due to generation of effective immunity following temporary lymphorestriction (the so called “Zitvogel effect”, reviewed in Benhar et al. J Clin Invest 2016).

However, several issues have to be addressed:

A possible approach to reach this goal is the investigation of the metabolic pathways exploited by the retrovirus to actively replicate and enter a latent state. This sentence is too vague, no clear that replication of virus (virions) lead to latent state, some infection in cell programmed to become long term memory with some stemness may drive the latency for the virus and not the virus inducing latency.

In the revised version of the manuscript, we have included data that strongly support a direct role of HIV-1 transcriptional state in determining the metabolic state of the host cell (please see reply to question n4 of Reviewer 2). However, we agree that metabolic changes of the cell itself, for example due to the reversal to a quiescent state, might also be responsible for the development of latency and this complementary scenario is now explicitly mentioned in the Discussion.

This individual had been treated with a combination ART and the NAD⁺ precursor nicotinamide, thus suggesting a possible contribution of glycolysis regulation in the therapeutic result obtained. Ref should be added Lebouché B, et al. . Impact of extended-release niacin on immune activation in HIV-infected immunological non-responders on effective antiretroviral therapy. HIV Res Clin Pract. 2021 Jan 6:1-9.

We thank this Referee for pointing out this reference, which has now been added in the description of the different types of metabolic approaches to an HIV cure.

Data presented are complex and combine immunology with biochemistry the focus should be on CD4 T cells, not sure that myeloid cell analyses and their heterogeneity and tissue dependence may be better described in another paper.

In line with this comment and with question n2 of the Referee, the main figures in the revised manuscript now only contain data derived from lymphoid cells. All data derived from other cell types are now presented in the Expanded View or Appendix material. In this regard, please note that the metabolomic data, previously available only for myeloid cells, have been now substituted with an entirely new dataset derived from lymphoid Jurkat T-cells, which allowed us to reach very similar conclusions and rendered the main message of the paper, *i.e.* the role of downregulated glycolysis as a specific marker of HIV-1 latency, more cohesive (please see Figure 2 of the revised manuscript above).

Educative table should be added with glycolysis as the pathway with NADH and the Krebs cycle to better understand study findings and their interactions.

We have now included schematic depictions of the metabolites of glycolysis as well as the intertwined Krebs cycle and pentose phosphate pathway, as suggested by the Reviewer (please see below as an example Figure 3 of the revised manuscript as well as Figure EV4 in the manuscript itself).

Figure 3 of the revised manuscript. Schematic depiction of glycolysis and related metabolic networks in latently HIV-1 infected Jurkat T-cells. The regulation of metabolic pathways was reconstructed based on the metabolomic data of latently infected Jurkat T-cells (2D10) as compared to their uninfected counterpart (E6). Enzymes are shown in italics. The figure includes the main energetic pathways described in the paper, as well as relevant connections with pathways that were significantly enriched in the analysis shown in Figure 2A (grey boxes). The red arrows indicate the proposed path of the glucose carbon chains in latently infected cells. Solid lines indicate direct connections. Dashed lines indicate indirect connections involving intermediate metabolites not shown in the figure. Networks were built using the Cytoscape software (<http://www.cytoscape.org>) and the Metscape plugin (<http://metscape.ncibi.org/tryplugin.html>) (Gao *et al.*, 2010) and adapted using Adobe Illustrator (v 16.03).

ACLY = ATP citrate lyase; ACO = aconitase; ADPGK = ADP dependent glucokinase; ALDO = fructose 1,6 bisphosphate aldolase; ENO1 = enolase 1; FBPI = fructose-bisphosphatase; PFK = phosphofructokinase 1; FH = fumarate hydratase; H6PD = hexose-6-phosphate dehydrogenase; HK1 = hexokinase 1; G6PD = glucose-6-phosphate dehydrogenase; GAPDH = glyceraldehyde 3-phosphate dehydrogenase; GCK = glucokinase; GPDH = glycerol 3-phosphate dehydrogenase; GPI = glucose-6-phosphate isomerase; KORA = ketoglutarate dehydrogenase A; IDH = isocitrate dehydrogenase; LDHAL6A = lactate dehydrogenase A like 6A; MDH1 = malate dehydrogenase 1; PCK1 phosphoenolpyruvate carboxykinase 1; PFK = phosphofructokinase; PGAM4 = phosphoglycerate mutase family member 4; PGD = 6-phosphogluconate dehydrogenase; PGK1 = phosphoglycerate kinase 1; PGLS = 6-phosphogluconolactonase; PGM1 = phosphoglucomutase 1; PKLR = pyruvate kinase L/R; PRPS1 = ribose phosphate pyrophosphokinase1; RPIA = ribose 5-phosphate isomerase; SDHA = succinate dehydrogenase complex flavoprotein subunit A; SUCD = succinate semialdehyde dehydrogenase; TKT = transketolase; TPI = triose phosphate isomerase.

In the introduction other metabolic pathway are also under investigation for HIV cure and should be shortly discussed with new reference.

A specific paragraph has now been added to the Introduction.

To limit the study to T cells and remove myeloid cells

As mentioned in the replies to questions n2 and n7, the main figures in this revised version now only contain data derived from lymphoid cells. Data from other cell types has been moved to the Expanded View or Appendix material, where we think it can be useful to pinpoint those metabolic changes that are conserved across latently infected cells of different lineages (*i.e.* glycolysis downregulation and compensatory activity of the pentose phosphate pathway).

To assess naïve vs memory cells separated by flow to decipher subset of CD4 T cells according to glycolysis status

Unfortunately, the separation of naïve and memory T-cells by flow cytometry can hardly be done in our model, because cells are first activated to allow HIV-1 infection. Once this occurs, cells typically lose the naïve phenotype. Moreover, measuring the whole glycolytic pathway by flow cytometry would be challenging. However, we agree with this Referee that, from the data that we presented, one might ascribe the different expression of glycolytic genes observed in infected and uninfected cells to a variable contribution of CD4⁺ T-cell subsets. To address this question, and the partially overlapping question n4 of the same Referee, we have now provided evidence that downregulation of the glycolytic pathway is a marker of latency independently of the cell subset considered (please see above Figure EV1 of the revised manuscript).

14th Jun 2021

Thank you for the submission of your revised manuscript to EMBO Molecular Medicine. We have now received the enclosed report from the two referees who were asked to re-assess it. As you will see below, the referees are now supportive and I am pleased to inform you that we will be able to accept your manuscript pending the following amendments:

1. In the main manuscript file, please do the following:

- remove the red color font.
- Author contribution: Virender Kumar Pal is listed as PKV - Please fix it.
- Reduce keyword number to 5.
- In Materials and Methods, include a statement that the experiments conformed to the principles set out in the WMA Declaration of Helsinki and the Department of Health and Human Services Belmont Report.

2. Checklist: both corresponding authors' names should be listed.

3. Fig EV1 contains two identical pages, please fix it.

4. Appendix:

- All Appendix Tables and Figures need to be merged together into a single pdf file called "Appendix". Provide a Table of Content on the 1st page.
- Appendix Table needs to be called out as Appendix Table S1 - NOT Appendix S1.
- All Appendix Figures need to be dropped down by one number because of the incorrect labelling of Appendix Table S1.
- Please double-check all Appendix Figure call-outs. Some callouts use the 'S' others don't, e.g., Appendix Figure S1 NOT Appendix figure 1.

5. Conflicts of interest: According to our editorial policy with regard to the "conflict of interest" (see below), the current statement suggests that you have no specific financial interest to declare - please confirm that.

'the journal requires authors of original research papers to declare any competing commercial interests in relation to the submitted work. It is difficult to specify a threshold at which a financial interest becomes significant, but as a practical guideline, we would suggest this to be any undeclared interest that could embarrass you were it to become publicly known.'

<https://www.embopress.org/page/journal/17574684/authorguide#conflictsofinterest>

Please also add 'The other authors declare no conflict of interests.'

6. For more information: There is space at the end of each article to list relevant web links for further consultation by our readers. Could you identify some relevant ones and provide such information as well? Some examples are patient associations, relevant databases, OMIM/proteins/genes links, author's websites, etc...

7. Data availability:

- Please note that the Data Availability Section is restricted to new primary data that are part of this study.

-GSE163405 and GSE163979 are not yet publicly available, please make sure that they will be accessible upon acceptance of the paper.

- please use the following format:

Data availability

- RNA-Seq data: Gene Expression Omnibus GSE46843

(<https://www.ncbi.nlm.nih.gov/geo/query/acc.cgi?acc=GSE46843>)

- [data type]: [name of the resource] [accession number/identifier/doi] ([URL or identifiers.org/DATABASE:ACCESSION])

8. We would also encourage you to include the source data for figure panels that show essential data. Numerical data should be provided as individual .xls or .csv files (including a tab describing the data). For blots or microscopy, uncropped images should be submitted (using a zip archive if multiple images need to be supplied for one panel). Additional information on source data and instruction on how to label the files are available at

<https://www.embopress.org/page/journal/17574684/authorguide#sourcedata>

9. synopsis image: The text becomes somewhat blurry when the image is adjusted to the required resolution (550 px width). Please provide a new image (PNG format) with clearer text (for instance, by increasing the text size). Please note that the resolution in which the image will be published is 550px width x 400-600 px height.

10. I have slightly modified the synopsis text. Please let me know if it is fine like this or if you would like to introduce future modifications. Please note that this would be the final version, and changes of synopsis text and image during proofing are usually not allowed.

The upregulation of glycolysis in activated cells favors HIV-1 infection and initial viral replication. This study discovers that, to transit into a latent form, which can shield the virus from immunity and antiretroviral drugs, HIV-1 needs to downregulate glycolysis.

- Restoration of glycolytic activity is required for HIV-1 reactivation from latency.
- Latently infected cells rely on pentose phosphate metabolism and its downstream effectors, i.e., the antioxidant glutathione and thioredoxin pathways, for their survival.
- Preferential targeting latently infected cells with drugs inhibiting thioredoxin and glutathione pathways leads to both HIV-1 reactivation from latency and death of infected cells.

11. Our data editors have seen the manuscript, and they have made some comments and suggestions that need to be addressed (see attached). Please send back a revised version (in track change mode), as we will need to go through the changes.

12. As part of the EMBO Publications transparent editorial process initiative (see our Editorial at <http://embomolmed.embopress.org/content/2/9/329>), EMBO Molecular Medicine will publish online a Review Process File (RPF) to accompany accepted manuscripts.

a. In the event of acceptance, this file will be published in conjunction with your paper and will

include the anonymous referee reports, your point-by-point response and all pertinent correspondence relating to the manuscript. Let us know if you do NOT agree with this.

I look forward to seeing a revised version of your manuscript as soon as possible.

Sincerely,
Jingyi

Jingyi Hou
Editor
EMBO Molecular Medicine

*** Instructions to submit your revised manuscript ***

To submit your manuscript, please follow this link:

Link Not Available

- 1) a .docx formatted version of the manuscript text (including Figure legends and tables)
- 2) Separate figure files*
- 3) supplemental information as Expanded View and/or Appendix. Please carefully check the authors guidelines for formatting Expanded view and Appendix figures and tables at <https://www.embopress.org/page/journal/17574684/authorguide#expandedview>
- 4) a letter INCLUDING the reviewer's reports and your detailed responses to their comments (as Word file).

5) The paper explained: EMBO Molecular Medicine articles are accompanied by a summary of the articles to emphasize the major findings in the paper and their medical implications for the non-specialist reader. Please provide a draft summary of your article highlighting

6) For more information: There is space at the end of each article to list relevant web links for further consultation by our readers. Could you identify some relevant ones and provide such information as well? Some examples are patient associations, relevant databases, OMIM/proteins/genes links, author's websites, etc...

7) Author contributions: the contribution of every author must be detailed in a separate section.

8) EMBO Molecular Medicine now requires a complete author checklist (<https://www.embopress.org/page/journal/17574684/authorguide>) to be submitted with all revised manuscripts. Please use the checklist as guideline for the sort of information we need WITHIN the manuscript. The checklist should only be filled with page numbers where the information can be found. This is particularly important for animal reporting, antibody dilutions (missing) and exact values and n that should be indicated instead of a range.

9) Every published paper now includes a 'Synopsis' to further enhance discoverability. Synopses are displayed on the journal webpage and are freely accessible to all readers. They include a short stand first (maximum of 300 characters, including space) as well as 2-5 one sentence bullet points that summarise the paper. Please write the bullet points to summarise the key NEW findings. They should be designed to be complementary to the abstract - i.e. not repeat the same text. We encourage inclusion of key acronyms and quantitative information (maximum of 30 words / bullet point). Please use the passive voice. Please attach these in a separate file or send them by email, we will incorporate them accordingly.

You are also welcome to suggest a striking image or visual abstract to illustrate your article. If you do please provide a jpeg file 550 px-wide x 400-px high.

10) A Conflict of Interest statement should be provided in the main text

11) Please note that we now mandate that all corresponding authors list an ORCID digital identifier. This takes <90 seconds to complete. We encourage all authors to supply an ORCID identifier, which will be linked to their name for unambiguous name identification.

Currently, our records indicate that the ORCID for your account is 0000-0003-0983-3693.

Please click the link below to modify this ORCID:
Link Not Available

12) The system will prompt you to fill in your funding and payment information. This will allow Wiley to send you a quote for the article processing charge (APC) in case of acceptance. This quote takes into account any reduction or fee waivers that you may be eligible for. Authors do not need to

pay any fees before their manuscript is accepted and transferred to our publisher.

Photos 400-800 DPI

*Additional important information regarding figures and illustrations can be found at <https://bit.ly/EMBOPressFigurePreparationGuideline>

The system will prompt you to fill in your funding and payment information. This will allow Wiley to send you a quote for the article processing charge (APC) in case of acceptance. This quote takes into account any reduction or fee waivers that you may be eligible for. Authors do not need to pay any fees before their manuscript is accepted and transferred to our publisher.

***** Reviewer's comments *****

Referee #2 (Comments on Novelty/Model System for Author):

N/A

Referee #2 (Remarks for Author):

The authors have adequately addressed the concerns raised in the initial review. I consider that the manuscript is now acceptable for publication.

Referee #3 (Comments on Novelty/Model System for Author):

Issues have been addressed

Referee #3 (Remarks for Author):

Issues have been addressed

The authors performed the requested editorial changes.

28th Jun 2021

We are pleased to inform you that your manuscript is accepted for publication and is now being sent to our publisher to be included in the next available issue of EMBO Molecular Medicine.

We would like to remind you that as part of the EMBO Publications transparent editorial process initiative, EMBO Molecular Medicine will publish a Review Process File online to accompany accepted manuscripts. If you do NOT want the file to be published or would like to exclude figures, please immediately inform the editorial office via e-mail.

Please read below for additional IMPORTANT information regarding your article, its publication and the production process.

Congratulations on your interesting work,

Jingyi

Jingyi Hou
Editor
EMBO Molecular Medicine

Follow us on Twitter @EmboMolMed
Sign up for eTOCs at embopress.org/alertsfeeds

***** Reviewer's comments *****

*** ** IMPORTANT INFORMATION *** **

SPEED OF PUBLICATION

The journal aims for rapid publication of papers, using using the advance online publication "Early View" to expedite the process: A properly copy-edited and formatted version will be published as "Early View" after the proofs have been corrected. Please help the Editors and publisher avoid delays by providing e-mail address(es), telephone and fax numbers at which author(s) can be contacted.

Should you be planning a Press Release on your article, please get in contact with embomolmed@wiley.com as early as possible, in order to coordinate publication and release dates.

LICENSE AND PAYMENT:

All articles published in EMBO Molecular Medicine are fully open access: immediately and freely

available to read, download and share.

EMBO Molecular Medicine charges an article processing charge (APC) to cover the publication costs. You, as the corresponding author for this manuscript, should have already received a quote with the article processing fee separately. Please let us know in case this quote has not been received.

Once your article is at Wiley for editorial production you will receive an email from Wiley's Author Services system, which will ask you to log in and will present you with the publication license form for completion. Within the same system the publication fee can be paid by credit card, an invoice, pro forma invoice or purchase order can be requested.

Payment of the publication charge and the signed Open Access Agreement form must be received before the article can be published online.

PROOFS

You will receive the proofs by e-mail approximately 2 weeks after all relevant files have been sent to our Production Office. Please return them within 48 hours and if there should be any problems, please contact the production office at embopressproduction@wiley.com.

Please inform us if there is likely to be any difficulty in reaching you at the above address at that time. Failure to meet our deadlines may result in a delay of publication.

All further communications concerning your paper proofs should quote reference number EMM-2020-13901-V3 and be directed to the production office at embopressproduction@wiley.com.

Thank you,

Jingyi Hou
Editor
EMBO Molecular Medicine

Corresponding Author Names: Iart Luca Shytaj, Andrea Savarino

Journal Submitted to: Embo Molecular Medicine

Manuscript Number: EMM-2020-13901-V2